# Thrombopoiesis is spatially regulated by the bone marrow vasculature

David Stegner [1], Judith M.M. van Eeuwijk[1], Oğuzhan Angay[2], Maximilian G. Gorelashvili[1], Daniela Semeniak[1], Jürgen Pinnecker[2], Patrick Schmithausen[2], Imke Meyer[3], Mike Friedrich[2], Sebastian Dütting[1], Christian Brede[4], Andreas Beilhack[4], Harald Schulze [1,3], Bernhard Nieswandt[1,2] & Katrin G. Heinze[2]

In mammals, megakaryocytes (MKs) in the bone marrow (BM) produce blood platelets, required for hemostasis and thrombosis. MKs originate from hematopoietic stem cells and are thought to migrate from an endosteal niche towards the vascular sinusoids during their maturation. Through imaging of MKs in the intact BM, here we show that MKs can be found within the entire BM, without a bias towards bone-distant regions. By combining in vivo two-photon microscopy and in situ light-sheet fluorescence microscopy with computational simulations, we reveal surprisingly slow MK migration, limited intervascular space, and a vessel-biased MK pool. These data challenge the current thrombopoiesis model of MK migration and support a modified model, where MKs at sinusoids are replenished by sinusoidal precursors rather than cells from a distant periostic niche. As MKs do not need to migrate to reach the vessel, therapies to increase MK numbers might be sufficient to raise platelet counts.

[1] Institute of Experimental Biomedicine, University Hospital Würzburg, Josef-Schneider-Str. 2, D15, 97080 Würzburg, Germany. [2] Rudolf Virchow Center for Experimental Biomedicine, University of Würzburg, Josef-Schneider-Str. 2, D15, 97080 Würzburg, Germany. [3] Laboratory of Pediatric Molecular Biology, Charité-University Hospital Berlin, Rudolf-Virchow-Klinikum, Augustenburger Platz 1, 13353 Berlin, Germany. [4] Department of Medicine II and Interdisciplinary Center for Clinical Research (IZKF), University Hospital Würzburg, Zinklesweg 10, 97078 Würzburg, Germany. David Stegner and Judith M. M. van Eeuwijk contributed equally to this work. Correspondence and requests for materials should be addressed to D.S. (email: stegner@virchow.uni-wuerzburg.de) or to K.G.H. (email: katrin.heinze@virchow.uni-wuerzburg.de)

Platelets play key roles in hemostasis and thrombosis and are the second most abundant cell type in the blood. Due to their short life span of only a few days, anuclear platelets are continuously replenished and thus provide a classic system to study hematopoiesis. The hematopoietic growth factor thrombopoietin (TPO) is the major cytokine triggering platelet production. TPO supports the self-renewal of hematopoietic stem cells (HSCs) and also induces transcription factors leading to the expression of proteins like CD42 (GPIb-V-IX complex) or CD41 (GPIIb) that commit HSCs to the platelet lineage[1, 2]. These committed precursor cells, designated megakaryocytes (MKs), then increase markedly in size and become polyploid. During their final maturation under the transcription factor NF-E2, MKs express the MK/platelet-specific tubulin isoform β1[3, 4]. Cytoplasmic MK-extensions called proplatelets pass through the endothelial barrier at bone marrow (BM) sinusoids—as recently suggested[5]—within the lungs, and are shed into the circulation. Each MK releases hundreds of virtually identical-sized platelets into the blood vessel[2, 6]. Under inflammatory conditions or acute platelet demand, platelet release also occurs via rupture of the mature MK membrane[7]. In both cases (pro-platelet formation and MK rupture), MKs must reside next to the vessel to release platelets into the bloodstream. According to the current model of megakaryopoiesis, blood cell precursors migrate from an endosteal niche towards the vessel sinusoids during maturation[1, 8–11]. This concept is primarily based on qualitative and quantitative evaluation of distinct progenitor cell populations present at distinct spatiotemporal niches. A seminal paper by Avecilla et al. has demonstrated that, while mice lacking TPO or its receptor c-Mpl have severely reduced platelet counts, the systemic application of the chemokines stromal cell-derived factor-1 (SDF1, CXCL-12) together with fibroblast growth factor 4 (FGF4) can transiently restore the number of peripheral platelets by directing MKs towards BM sinusoids[1, 8]. Interestingly, in contrast to the MK maturation model, Junt and colleagues observed by intravital two-photon microscopy (2P-IVM) that MKs barely migrate and are mostly found in close proximity to blood vessels[6]. Unfortunately, this previous study has assessed only a relatively small number of MKs, as the field of view in 2P-IVM is limited and due to the CD41-YFP reporter mice used. In these mice only one third of MKs become fluorescently labeled due to unexplained reduced penetrance of the transgene, while the CD41/61 (GPIIb/IIIa) expression is reduced, due to the heterozygous CD41-knockout in these animals[6, 12]. Thus, the authors mostly used TPO-treated mice to enhance the number of visible cells. So far, the discrepancy between the current model of megakaryopoiesis and the in vivo data shown by Junt et al. has not been reconciled.

As a result of recent developments in imaging techniques, we were able to analyze the distribution of MKs within the bone marrow by combining different in vitro and in vivo imaging techniques with computational simulations. We provide independent lines of evidence that challenge the directed MK migration model and provide a modified model, where MKs at sinusoids are replenished by precursors originating from a sinusoidal niche rather than from a periostic niche.

## Results

**Most MKs are residing directly at the vessels**. We performed in vivo antibody labeling targeting the MK-specific and platelet-specific von Willebrand factor receptor subunit GPIX (CD42a) to overcome the limitations of the CD41-YFP mouse and to stain the entire MK population in the mouse BM, without affecting platelet production (Supplementary Figs 1A–D). This labeling scheme also obviated the necessity for TPO-treatment to increase the number of labeled MKs. Since platelets are continuously produced, we analyzed MK migration under steady-state conditions, visualizing the BM in the mouse skull using 2P-IVM. We tracked single MKs (n = 54) over time at a frame rate of 1f/min and discovered that the cells were sessile (Fig. 1a–c, Supplementary Movie 1). MKs had an instantaneous velocity of 0.38 ± 0.47 μm/min (median ± SD) and displayed either just a wobbling or shifted their center of mass during proplatelet formation at BM sinusoids, in line with previous reports[6]. We then assessed MK mobility by analyzing MK tracks based on an anomalous diffusion model. The mean squared displacement (MSD = $< r^2(t) > = 4 K_\alpha t^\alpha$) of MKs was calculated for different time scales $t$ as the MSD can be directly linked to (the time scale-dependent) diffusion coefficient $K_\alpha$ and the anomalous diffusion exponent $\alpha$. For MK migration, we found low values (median ± SD) for the MSD (0.35 ± 0.41 μm²) and $\alpha$ (0.65 ± 0.22), indicating a slow, subdiffusion-like movement in a highly crowded environment (Fig. 1c), further demonstrating hindered movement of MKs in the BM.

As most studies of murine hematopoiesis have been performed on long bones and 2P-IVM approach in the cranial BM has a limited field of view, we next performed immunohistochemical analysis of cryo-sections[13] of murine femora (Fig. 2a) and sterna (Fig. 2b). Under steady-state conditions, we found no regions lacking cells of the megakaryocytic lineage (CD41-positive). Surprisingly, MKs were not only found near the sinusoids (labeled with anti-CD105 or anti-CD31 antibodies), but also in

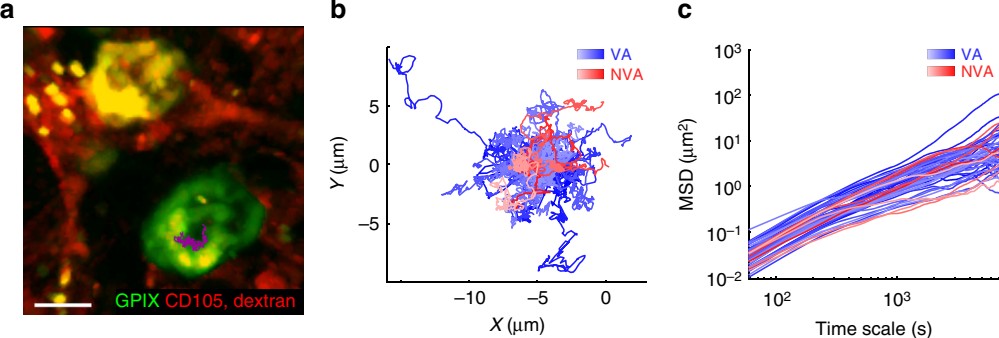

**Fig. 1** MK tracking reveals low motility of MKs in vivo. **a** A representative centroid track (*magenta*) from a MK (*green*, scale bar 15 μm) monitored for over 3 h. Frame rate: 1 f/min **b** Trajectories of individual MKs (n = 54 from six different mice) over a time period of 3 h; individual trajectories of MKs are represented by different *blue* (vessel-associated, n = 46) and *red* (non-vessel associated, n = 8) shadings. Frame rate: 1 f/min **c** MSD analysis of individual MKs (n = 54) for different time scales. Corresponding α coefficients can be derived from slopes in the log-log-plot; MSD curves of individual MKs are represented by different *blue* (vessel-associated, n = 46) and *red* (non-vessel associated, n = 8) shadings

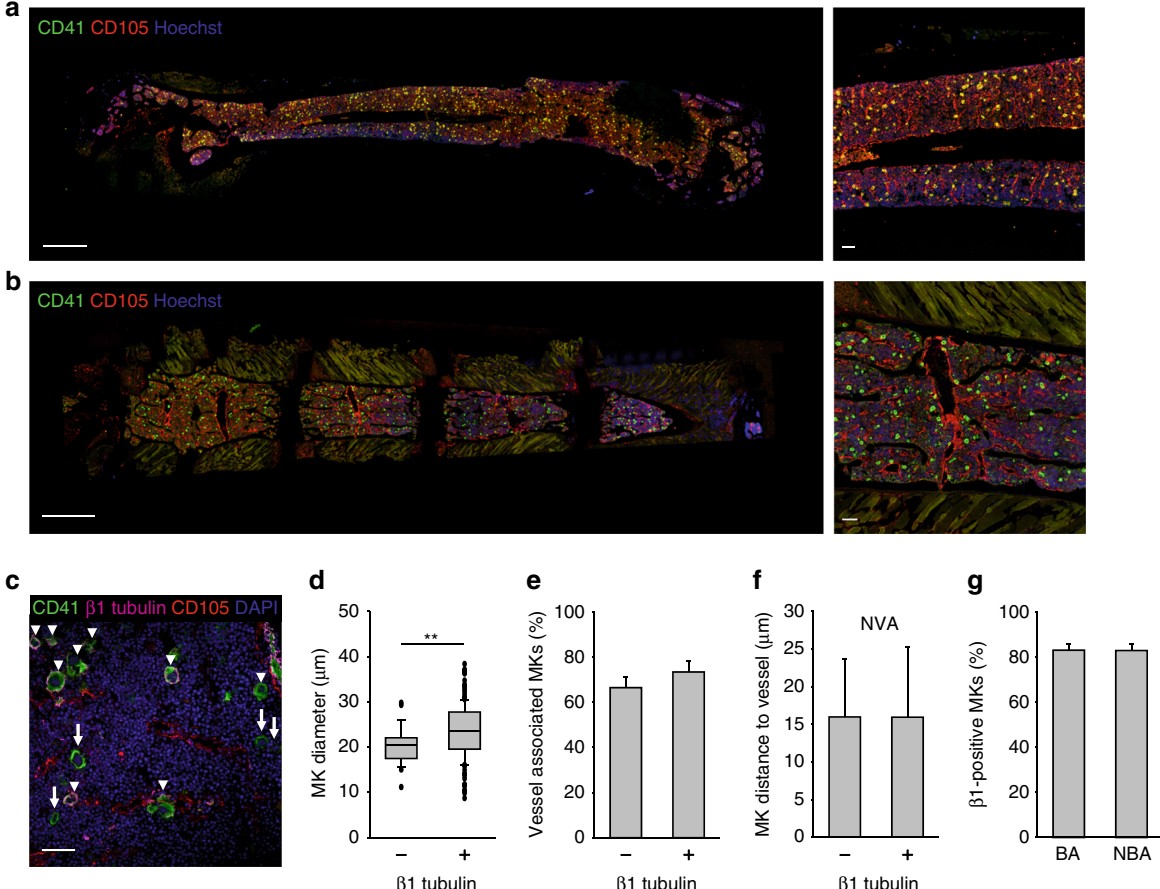

**Fig. 2** MKs are largely sessile and distributed within the BM. Distribution of megakaryocytes (MKs, CD41$^+$, *green*) in cryo-sections of femur **a** and sternum **b** showing vessels (CD105$^+$, *red*) and nuclei (Hoechst33258, *blue*, Scale bar: 1 mm (*left panels*), 100 μm (*right panels*). **c** MKs were counter-stained for β1-tubulin (*magenta*, indicated with *arrowheads*—in contrast to β1$^-$ MKs which are indicated with *arrows*); scale bar 20 μm. These mature MKs were larger **d**, but distributed similarly as depicted by the percentage of vessel-associated MKs **e** and the distance to vessels of non-vessel associated (NVA) MKs **f**. **g** The percentage of β1-tubulin positive CD41-cells within 100 μm distance from the bone cortex (BA bone-associated) and more distant MKs (NBA non-bone associated) was similar. Bar graphs represent mean ± SD; n = 4. **p < 0.01. There are no significant differences in **e**–**g** (p > 0.05; Mann–Whitney U test)

close contact to endosteal zones (indicated by *dark regions surrounding the BM*, Fig. 2a, b). MK diameters were comparable in sternum and femur (22.19 ± 0.14 μm vs. 22.57 ± 1.44 μm) and in both, the majority of MKs (68.2 ± 6.9% vs. 73.1 ± 5.2%) was found in direct contact with the vessel (distance < 2 μm), which is in agreement with previous studies[6, 14]. Of note, consecutive cryo-sections confirmed that more than 99% of CD41-positive MKs were also positive for GPIX (CD42a) (Supplementary Fig. 1A). Both groups also showed identical MK diameters and percentage of vessel association. We next examined the localization of the MK-specific tubulin isoform β1 as a marker for mature MKs[4] (Fig. 2c). As expected, mature β1-positive MKs were larger than immature β1-negative CD41-positive cells (23.53 ± 6.23 μm vs. 20.19 ± 4.24 μm, Fig. 2d). Surprisingly, however, maturation was not accompanied by altered MK distribution, as the majority of both β1-tubulin negative and positive cells were in direct contact with the vessel (66.9 ± 4.5 vs. 73.9 ± 4.9%; Fig. 2e). In addition, vessel-MK distances of non-vessel associated (NVA) MKs were indistinguishable for β1-tubulin negative and positive cells (Fig. 2f). Next, we assessed the percentage of β1-tubulin positive CD41-positive cells within 100 μm distance from the bone cortex, compared to more distant MKs. Interestingly, the percentage of β1-tubulin positive cells among all MKs was identical between the 'bone-associated' and more distant MK population (83.0 ± 2.8% vs. 82.7 ± 2.6%;

Fig. 2g). Thus, the majority of MKs was found in close proximity to the sinusoids, independently of their size or maturation state.

**MKs are distributed throughout the BM.** To assess the MK distribution in its intact 3D environment, we utilized light-sheet fluorescence microscopy (LSFM) to examine complete and intact bones (Supplementary Movie 2, Supplementary Fig. 2A, B). We first established a clearing protocol, leading to optically transparent bones (Supplementary Fig. 2C, D)[15–18]. To avoid staining artifacts due to insufficient antibody penetration in situ, we performed in vivo labeling using an anti-GPIX antibody derivative[19, 20]. LSFM transversal imaging of the cleared and intact femora and sterna allowed visualization of MKs with a resolution of less than 2 μm, which was sufficient to identify proplatelets extended by MKs (Supplementary Fig. 2B, Supplementary Movie 3 and 4). 3D reconstructions of LSFM data revealed a surprisingly dense blood vessel network with an average vessel-to-vessel distance of only 43.18 ± 12.72 μm in the sternum (Fig. 3a). The mean MK diameter was determined as 20.36 ± 8.19 μm (Fig. 3a), comparable to the values obtained with cryo-sections. Thus, the spatial distribution of MKs is highly restricted by the vasculature as the vessel-to-vessel distance equals the diameter of two MKs. In line with this, we found that 79.3 ± 5.1% of MKs localized adjacent to the blood vessels

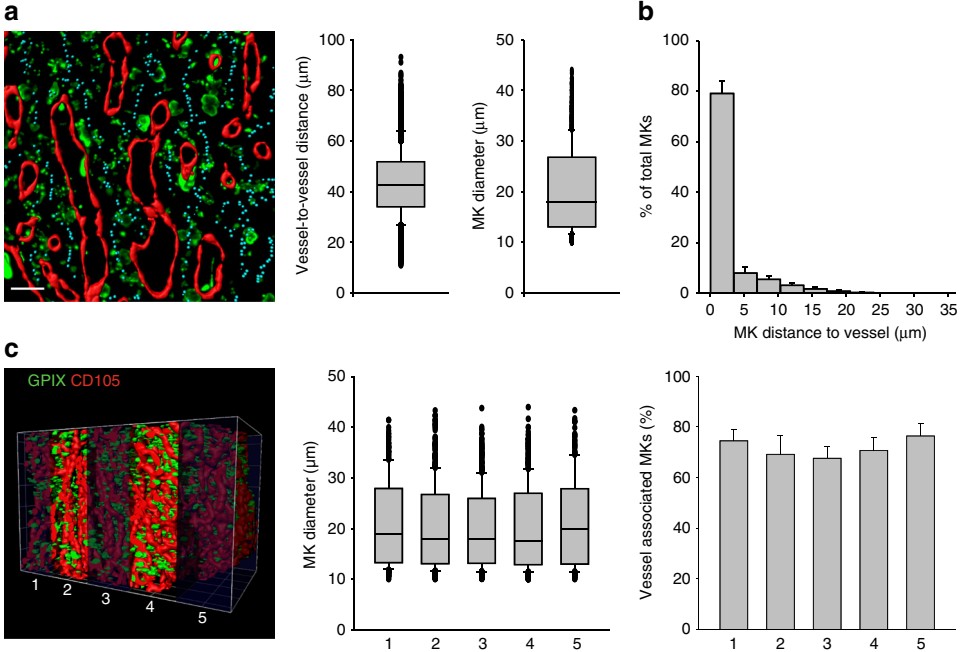

**Fig. 3** The intervascular space limits MK distribution within the BM. **a** Reconstruction of sternal BM revealed a dense blood vessel network (*red*, CD105) with limited space for MKs (*green*, GPIX) shown by vessel-to-vessel distances and average MK diameter (*n* = 5 mice). *Cyan* dots represent the center (maximal distance) between two adjacent vessels. Scale bar 50 μm. **b** Most MKs are localized adjacent to sternal sinusoids and **c** MK diameters and MK distribution, shown by the percentage of vessel associated MKs, were comparable throughout the whole sternum in five different sections of the reconstructed BM. Bar graphs represent mean ± SD; *n* = 5; grid square = 100 μm. There are no significant differences in **c** (*p* > 0.05; Mann–Whitney *U* test)

(Fig. 3b). This value is higher than those obtained by analyzing cryo-sections, likely due to vessels which are missed as they are located above or beneath the analyzed plane of the bone section. Analyzing the 3D LSFM data sets by virtual slicing revealed that the 3D information significantly improves quantitative cell analysis, as it reduces the risk of overrating cell distances due to 'off-center' cell slicing and other well-known cutting artifacts. Equally important, LSFM enabled us to analyze 30-fold more MKs per bone sample compared to conventional sections. Thus, we can demonstrate that within an entire bone the pool of MKs is distributed throughout the microvessel network. Analysis of five equally sized sub-stacks of the complete 3D stack of an intact bone revealed identical values in all MK parameters (Fig. 3c), further supporting the hypothesis that the entire BM contains MKs.

Next, we assessed whether the BM is structured differently between different types of bone. To this end, we confirmed that the bone-like structures at the outer linings of femur or sternum sections are positive for collagen I[21] (Supplementary Fig. 3A), and partly positive for osteocalcin (Supplementary Fig. 3B). We noted that in our LSFM data these structures displayed a characteristic auto-fluorescence signal (Supplementary Fig. 3C), enabling segmentation of bone structures from our LSFM data (Fig. 4a, Supplementary Fig. 4A, B). We determined the bone-to-bone and vessel-to-vessel distances of femur diaphysis (Supplementary Movie 5), femur epiphysis (Supplementary Movies 6 and 7), sternum (Supplementary Movie 8), and cranial BM (Supplementary Movie 9). Of note, while bone-to-bone distances differed between femur diaphysis and the other three types of bone (Fig. 4b), vessel-to-vessel vessel, as well as MK-to-vessel distances were indistinguishable (Fig. 4c). Consequently, the majority of MKs was vessel-associated in all four types of BM (Fig. 4d). As the 'classical concept' of megakaryopoiesis suggests a migration from the bone associated (BA) endosteal niche towards the vasculature during MK maturation, we compared

bone-associated; less than 100 μm distance from the bone cortex) and non-bone-associated (NBA) MKs (Fig. 4e) in femur diaphysis (Fig. 4f) and sternum (Fig. 4g). Notably, in either case BA and NBA MK parameters were indistinguishable (Fig. 4f, g), corroborating our finding that we could not detect any MK-deficient areas within the BM.

**Platelet depletion does not affect megakaryocyte motility**. Having established assays for MK motility and 3D distribution, we next tested the effects of increased platelet consumption. We depleted platelets using anti-GPIbα antibodies that cause rapid Fc-independent platelet clearance[19]. This antibody treatment resulted in severe thrombocytopenia (virtually no circulating platelets on d1; 10.4 ± 8.7% of the respective unaffected values on d3, restored platelet counts on d7, Fig. 5a). Based on the kinetics of the platelet turnover, we analyzed the BM on d3.5 post application 2P-IVM, a time point at which replenishing platelet production was expected to have reached its maximum. Interestingly, despite the severe thrombocytopenia, we found no evidence for any MK migration under these conditions, even for observation periods of several hours. MKs displayed an instantaneous velocity of 0.02 ± 0.06 μm/s (median ± SD) and MK tracks (Fig. 5b, c, Supplementary Movie 10) resembled those of steady-state conditions (Fig. 1b). Likewise, MSD values (0.40 ± 0.39 μm²) and α values (0.45 ± 0.18) were low, indicating that despite increased platelet turnover, MKs still displayed a slow and subdiffusion-like movement in a highly crowded environment.

To expand our observations to a larger field of view, we performed LSFM analyses of sterna at multiple time points post platelet depletion. Notably, we did not detect any signs of MK loss due to rupture and the main MK parameters remained grossly unaltered. We did observe a mild increase in total MK volume at d3-5 after platelet depletion, resulting in an increased fraction of

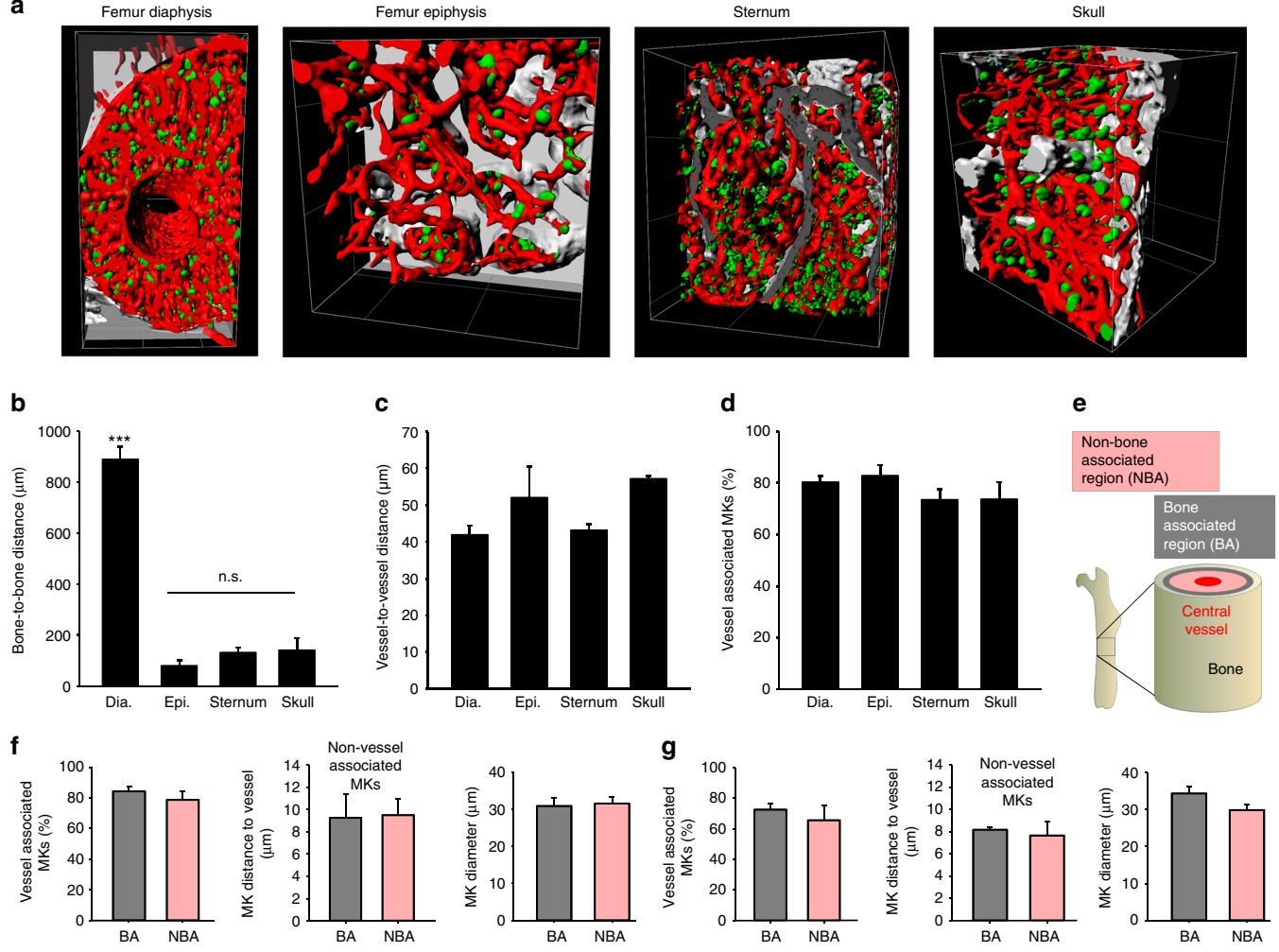

**Fig. 4** The inter-bone space is fully vascularized and contains MKs. **a** Reconstruction of BM along with bone structures for different types of bone revealed that the entire BM contains MKs (*green*, GPIX) and blood vessels (*red*, CD105). Grid square = 200 μm. **b** Quantitative analyses revealed that bone-to-bone distances differ between femur diaphysis (dia.) on the one hand and femur epiphysis (epi.), sternum, skull on the other hand. However, the vessel-to-vessel distances **c** and the percentage of vessel associated MKs **d** are comparable. MKs were grouped in bone associated (BA) or non-bone associated (NBA, more than 100 μm distance from the bone cortex) as depicted schematically **e**. A subsequent analysis of MK parameters for femur **f** or sternum **g** revealed that the percentage of vessel-associated MKs, the MK distance to the vessel of non-vessel-associated MKs or MK diameters were similar between BA and NBA MKs in both types of bone. Bar graphs represent mean ± SD; $n = 4$; ***$p < 0.001$ (Mann–Whitney $U$ test)

vessel-associated MKs as the "vessel-distant" space is smaller for bigger cells (Fig. 5d–f). The LSFM-data of the platelet-depleted mice indicate that even upon high platelet demand, the MK distribution remains overall unaltered, which stands in clear contrast to the idea that immature MKs migrate toward vessels.

Next, we assessed the DNA replication rate in vessel-associated and vessel-distant MKs as a readout for MK (endo-)replicatory activity. We used the nucleotide derivative 5-ethynyl-2'-deoxyuridine (EdU) that becomes incorporated into DNA of proliferating cells during DNA replication with a limited toxicity[22, 23]. In contrast to in situ DNA stainings, EdU labeling provides 'temporal' information, as it labels only DNA which had been synthetized in the presence of EdU. Maturing MKs undergo endoreplication, which would result in highly EdU-labeled cells. Mice were fed daily with EdU for 10 days in order to obtain levels with sufficient bioavailability for cellular uptake, DNA incorporation and reliable detection afterwards (Supplementary Fig. 5). Following platelet depletion or control antibody injection, EdU feeding was continued and sectioned bones were stained at the indicated time points to assess the

amount of EdU incorporation into MKs (Fig. 5g). As MKs have a range of ploidy grades and can undergo further rounds of endoreplication, they are expected to have achieved incorporation of the trackable nucleotide, depending on the MK ploidy level at start and the number of endoreplication cycles undergone in sufficient bioavailability. We classified MKs into the three categories: "no", "weak", and "strong", based on the staining intensity in the nucleus. As expected, the staining intensity increased over time (Fig. 5g). Notably, highly EdU-positive MKs were predominantly found in close proximity to the vasculature indicating that MK endoreplication occurs mostly directly at sinusoids. In mice treated with the depletion antibody, we found more EdU positive MKs on day 3 as compared to control animals, reflecting that the MK endoreplication rate was increased due to elevated platelet demand. However, at later time points the system 'bounced back' to 'normal' as the fraction of EdU-negative (= non-DNA replicating MKs) was similar between platelet-depleted and unchallenged mice on day 5 and 7 (Fig. 5g). Likewise, the fraction of strongly stained MK fraction was comparable between the two groups. These results further demonstrated the robustness of the MK distribution and

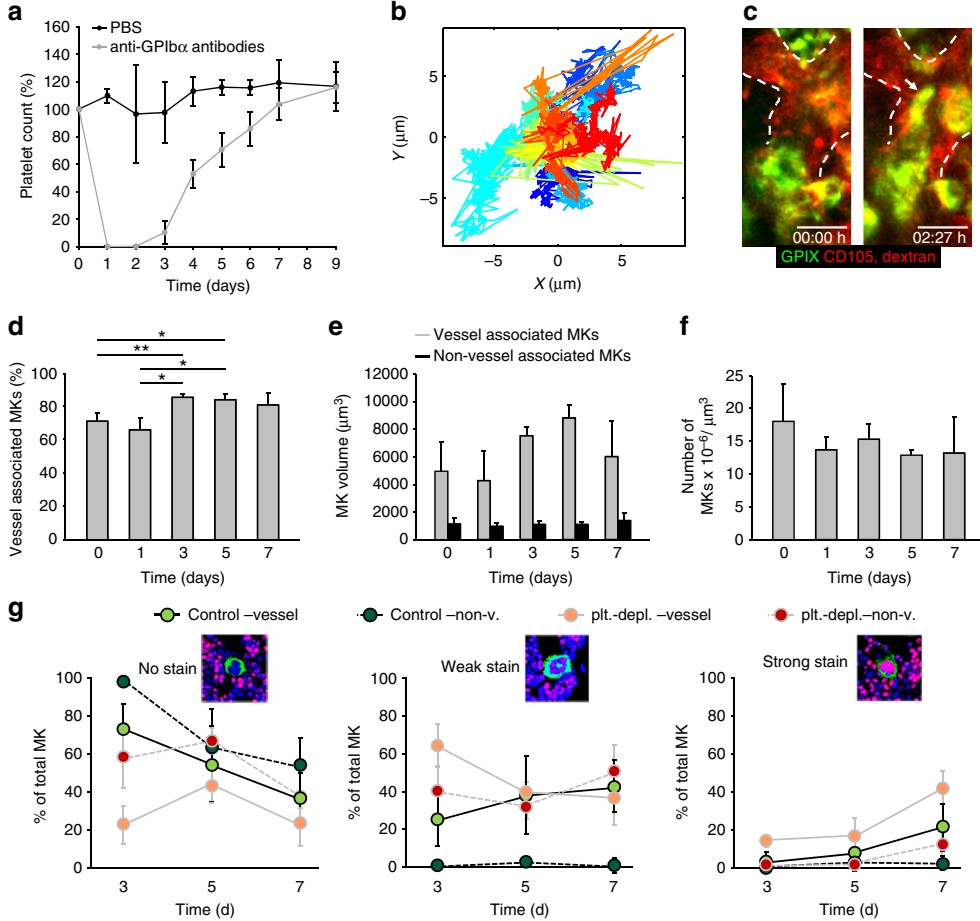

**Fig. 5** Platelet depletion does not affect MK distribution or migration within the BM. **a** Mice displayed severe, but reversible thrombocytopenia after platelets were depleted by anti-GPIbα antibodies (five mice per group). **b** Trajectories of individual MKs in cranial BM on d3.5 after platelet depletion; each *color* represents one MK track. **c** A representative MK that remained sessile for 2.5 h before starting to form proplatelets in a vessel (*dashed line*) (see also Supplementary Movie 10; scale bar 20 μm). **d–f** Sterna were analyzed by LSFM on day 1, 3, 5, and 7 following platelet depletion (five mice per day). Bar graphs represent mean ± SD. Increase in vessel associated MKs upon platelet depletion **d** correlates with an increase in MK-volume of vessel associated (*gray bars*, **e**) but not of non-vessel associated MKs (*black bars*, **e**). **f** Total MK numbers remain unaltered. Bar graphs represent mean ± SD. *$p < 0.05$, **$p < 0.01$ (Mann–Whitney $U$ test). There are no significant differences in **e** or **f** ($p > 0.05$; one-way ANOVA). **g** DNA synthesis rate in MKs in vivo after platelet depletion (five mice per day). Femur bones were sectioned and stained for EdU (*red*), which is incorporated into nuclei (*blue*). MKs are stained with anti-CD41 antibody (*green*). EdU incorporation was detected in vessel-associated (vessel) and non-vessel-associated (non-v.) MKs at the indicated time points in platelet depleted (plt-depl) and control injected mice

thrombopoiesis even under conditions of high platelet production.

**Short-term CXCR4 inhibition does not affect MK distribution.** The chemokine receptor CXCR4 has been shown to modulate MK transmigration through an endothelial layer in vitro[24–26] and its ligand SDF1 induces MK migration in vitro[27]. CXCR4 also contributes to platelet production, especially in the absence of the TPO-c-Mpl axis[8]. Therefore, we blocked CXCR4 in wild type mice using Plerixafor (AMD3100) and determined the effects on MK localization after 24 h. Treatment with Plerixafor resulted in elevated HSC counts (Sca-1+, c-kit+, Lin− cells) in the peripheral blood after 24 h (290 ± 56% of control counts), confirming previous results[28]. Platelet counts were slightly decreased 1 h after the treatment (768 ± 72 vs. 977 ± 56 platelets/nl blood), but indistinguishable from those of control mice after 24 h (987 ± 185 platelets/nl blood). However, CXCR4 blockade did not modulate MK localization or volume (Fig. 6a–c). Similar results were obtained in mice receiving the CXCR4 ligand SDF1 24 h before analysis (Fig. 6a–c), indicating that, despite the relevance of the

SDF1-CXCR4 axis for HSC mobilization[29], it appears to be negligible for MK (re)localization.

Integrin signaling is a central step in mediating cell adhesion and migration[30, 31]. Talin1 is a critical mediator of integrin activation and outside-in signaling[32, 33]. Consequently, we assumed that talin1-deficiency could affect MK migration and result in altered MK localization, despite the fact that MK/platelet-specific talin-deficient mice (*Tln^{fl/fl, Pf4-Cre}*) have unaltered platelet counts[32]. To assess this directly, BM sections of *Tln^{fl/fl, Pf4-Cre}* mice[32] were assessed (Fig. 6d). Surprisingly, the percentage of vessel associated MKs (Fig. 6e) and the MK-vessel distance of the NVA MKs (Fig. 6f) were indistinguishable between talin-deficient animals and littermate controls. These data indicate that classical integrin-dependent cell migration is not required during megakaryopoiesis.

**Simulations imply a vessel-biased MK distribution.** The high numbers of MK and their limited migratory capacity led us to hypothesize that MKs are randomly distributed throughout the BM. To test this hypothesis, we made use of computer simulations of possible patterns of MK distribution in their native

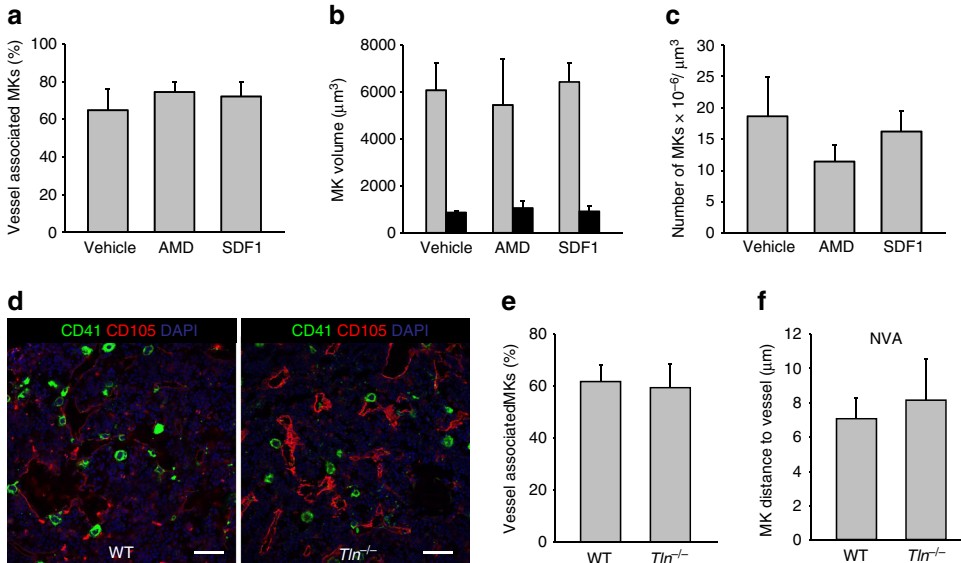

**Fig. 6** CXCR4-blockade or talin-deficiency does not affect MK distribution within the BM. Treatment with 5 mg/kg bodyweight AMD3100 (AMD) or 16 μg/kg bodyweight SDF1 did not modify MK localization **a**, volume **b** or numbers (four mice per group) (**c**). **d**, **e** Talin-deficiency ($Tln^{-/-}$) did not affect MK distribution (Scale bar 50 μm) or MK to vessel distances of non-vessel associated (NVA) MKs (**f**) ($n = 5$ for $Tln^{-/-}$, $n = 4$ for WT). Bar graphs represent mean ± SD. There are no significant differences ($p > 0.05$; one-way ANOVA)

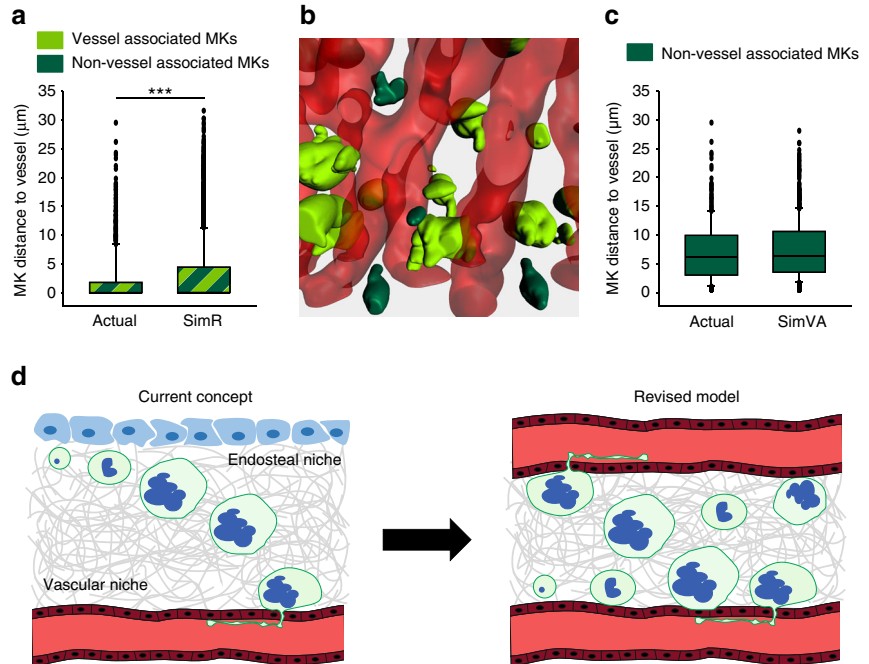

**Fig. 7** MKs display a vessel-biased random distribution. MK distribution was simulated ($n = 6$ simulations) using blood vessels (*red*) and MKs (*green*) derived from sternal BM imaging data. **a** The average distance of random distributed MKs to the vessel (SimR) is increased compared to actual data. ***$p < 0.001$. **b** Simulated vessel-associated MKs are depicted in *light green*, non-vessel-associated MKs are depicted in *dark green*. **c** Random simulation of only the non-vessel associated MK population results in average distances comparable to those obtained by 3D imaging data. **d** According to the current model of megakaryopoiesis (*left panel*), blood cell precursors migrate from the endosteal niche towards the vessel sinusoids during maturation. Our data support a revised model (*right panel*) where MKs reside directly at the sinusoids and are replenished by precursors originating from the sinusoidal niche rather than a periostic niche

3D environment. For modeling, the morphology and size of the MKs, as well as the BM vasculature, were derived from isosurfaces of experimental data, thus accurately reflecting the physiological situation spatially. The MK-like objects were randomly placed into the BM, however a fully random MK distribution (SimR) did not reflect the distribution from our 3D imaging results (Fig. 7a). The MK population in vivo is not randomly distributed,

but displays a vessel-biased distribution (Fig. 7a). The SimR simulation predicts significantly larger MK-vessel distances and a lower percentage of vessel-associated MKs (62.8 ± 1.9% compared to 70.1% in experimental data). Next, in a modified simulation, we only considered the MK subpopulation of the NVA MKs (SimVA, *dark green* in Fig. 7b) to be random. To this end, the BM vasculature with its vessel associated MKs was used

as a space restricting scaffold and 30% of the MKs were randomly seeded into the remaining intervascular region. Under these conditions, the model fully matched the distribution of NVA MKs from the 3D imaging data (Fig. 7c) and validates our previous conclusion that MKs do not migrate within the BM: If migration was required to reach the vessel, "free" (NVA) MKs would show an increasing density gradient towards the vessels. In contrast, our results emphasize a random distribution of both immature and mature "free" MKs throughout the BM. Thus, the computer simulation data and the statistical analysis of MK-vessel distances in the whole intact bone suggest that the majority of MKs reside directly at the vasculature (Fig. 7d), indicating that they are replenished by progenitors in close spatial proximity.

## Discussion

Our whole-organ imaging approach revealed that the entire inter-bone space is filled with BM that is fully vascularized, leaving no space for vessel-distant-niches. The combination of our experimental and modeled data strongly requests and supports a revised model of megakaryopoiesis, with the majority of MKs residing directly at the sinusoids (Fig. 7d), independently of their distance to the bone cortex (Fig. 4). Thus, vasculature seems to dictate the distribution of the vast majority of MKs, namely those in direct contact to the vessels. In contrast, NVA MKs are randomly distributed across the BM with negligible mobility. Moreover, there are no differences in the distribution of 'young' and mature MKs (Fig. 2g). This clearly contradicts the current idea of cellular migration during megakaryopoiesis.

Notably, the current model is based primarily on the seminal paper of Avecilla et al., who revealed that a co-injection of SDF1 and FGF4 into c-MPL-deficient mice transiently increases platelet counts[8]. The increased MK numbers at the sinusoids after SDF1-injection have been interpreted as result of MK migration towards the vessel. However, an increase of MK numbers—due to accelerated HSC differentiation—would inevitably result in an increase of MKs with vessel contact, even if the relative MK distribution remained unaltered. As HSCs and other progenitor cells are also affected by the CXCR4–SDF1 axis[34], it is possible that this initial study[8], which used relatively long observation periods after SDF1 treatment, was confounded by effects acting rather indirectly on the spatiotemporal MK distribution.

The current model of MK migration is also supported by in vitro data[24, 26, 27]. However, two of these in vitro studies assessed MK transmigration through an endothelial layer[24, 26]. Thus, CXCR4 might be critical to facilitate the last step of MKs and to some extent polarize them, so that they adhere to the endothelial layer (as it has been described in vitro by[24]) before they start forming proplatelets. However, to the best of our knowledge, WHIM syndrome (warts, hypogammaglobulinemia, infections, and myelokathexis syndrome), which is at least in some patients caused by hyperreactive CXCR4 signaling[35] is not associated with altered platelet counts. Thus, it may as well be that CXCR4 signaling is not required for thrombopoiesis. Mazharian et al. assessed MK migration using cell tracking and visualizing cell movement in vitro, albeit in the absence of BM environment[27]. They revealed that SDF1-triggered MK migration depends on the Src-Syk-PLCγ2 axis. However, platelet counts are not affected by MK and platelet specific Syk-deficiency[36]. Of note, deficiency of talin1 or kindlin-3, both critical mediators of integrin activation and cell migration, did not affect platelet counts either[32, 37]. More importantly, we did not observe any effects of talin-deficiency on MK distribution (Fig. 6). Likewise, leukocyte adhesion deficiency, type III patients, who have loss-of-function mutations in FERMT3 (the gene encoding for kindlin-3) do not suffer from thrombocytopenia. Independently of the

relevance of SDF1-CXCR4 signaling in vitro, our study clearly demonstrates that MKs do not need to migrate to the vessel due to the low vessel-to-vessel distances. One reason appears to be the dense BM vasculature restricting the extraluminal space in vivo, thereby precluding the need for MK migration, as growing MKs will reach a vessel with minimal movement. In addition, our 2P-IVM data, with up to 4-h long recordings, show that MKs barely move. The vessel-associated localization of MKs enables a rapid response under conditions of increased platelet need. After platelet depletion we did not observe MK rupture, indicating that only under inflammatory conditions[7] platelet generation may occur without proplatelet formation. Interestingly, platelet depletion did neither affect MK migration (Fig. 5b, c) nor the MK distribution in the BM (Fig. 5d–f). Such overall unaltered MK distribution, even under conditions of elevated platelet production, provides indirect evidence for the existence of a key regulatory mechanism that places the origin of the vast majority of MKs in closest vicinity to the vessels. In support of this, we observed a similar distribution of 'young' (β1-tubulin negative) vs. mature MKs (Fig. 2g). This allows platelet counts to recover quickly after depletion. Even if the exact mechanism remains to be elucidated, the findings here unveil that it is crucial to pay particular attention to the vascular niche when it comes to the origin of MKs. In contrast, the 'real' endosteal zone, which lacks vascularization, is restricted to the outer part of the bone and basically cell-free (Figs. 2 and 4, Supplementary Fig. 3). When assessing MK distribution close to these 'outer' zones, we did not observe any differences compared to the MK distribution in the 'inner', clearly not endosteal, region (Figs. 2 and 4). Thus, we have not depicted the endosteal niche in our revised model (Fig. 7d). Our data rather support the concept of functional endosteal and vascular niches within the BM which are not completely separable[38]. This is in agreement with previous studies indicating perivascular 'niches' for HSCs within the BM[39–41], and is further supported by a recent report demonstrating that non-dividing stem cells mainly localize at a perisinusoidal site[42].

Besides their 'classical' niche within the BM, murine MKs can also be found in the spleen. Platelet production has also been described within the capillary beds of the lungs[5, 43, 44]. However, after analyzing more than 230 MKs in total and more than 24 h of 2P-IVM video recordings we observed almost no MKs within the BM vasculature (0.87% of MKs). Using LSFM, which allowed analysis of high sample numbers, MKs were rarely found within the vasculature (2.0 ± 1.6% of all MKs). Thus, it appears unlikely that the translocation of BM-derived mature MKs into different organs would be a dominant phenomenon.

A vessel-biased MK distribution is critical to enable efficient proplatelet formation upon demand. However, this MK distribution appears to be the consequence of the dense microvasculature and presumably perisinusoidal HSCs[42]. Consequently, it appears likely that MK precursor cells reside in close proximity to the vasculature, allowing the rapid generation of new MKs upon platelet demand (Fig. 3). In addition, the vessel-biased distribution of MKs could be supported by MK polarization (e.g., via 'directed growth'). Polarization may be triggered by mechanical cues, such as adhesion to endothelial cells expressing vascular cell-adhesion molecule-1[8] or by environmental soluble factors such as SDF1 or sphingosine-1-phosphate[8, 14]. With that in mind, MK positioning could affect platelet production rather than megakaryopoiesis. However, we cannot rule out the possibility that two pools of MKs exist: one pool that resides directly at the vessel to readily produce platelets (e.g., on demand), whereas a second, largely quiescent pool of MKs or (potentially) megakaryocytic progenitor cells, resides more distant from the vasculature. The hypothesis of two rather distinct MK pools is supported by our EdU-experiment (Fig. 5g).

This experiment allows to distinguish between two models: A 'serial tube' model where all mature MKs are used for platelet release before new precursors differentiate and take up EdU (first in-first out, all mature MKs would be EdU-positive) or a 'dual MK pool' model in which mature EdU-negative MKs are still present. In fact, our results show a substantial fraction of MKs to be EdU-negative (even for an extended EdU-feeding period and increased platelet demand) so that the 'dual MK pool' model is the more likely one. A similar concept has been proposed for HSCs[40, 45], which reside either directly at the blood vessels or at the endosteal niche[39–42]. The combination of previous reports on MK-biased HSCs that regulate HSC activity[46] and the finding that MK-secreted factors keep HSCs quiescent[47, 48] raises the intriguing possibility that MKs regulate the MK-biased pool of HSCs.

These findings have immediate translational implications: On the one hand, the vessel-biased MK distribution indicates, that promoting megakaryopoiesis (e.g., by differentiation-promoting chemokines) might be sufficient to rapidly enhance platelet production, as MKs do not need to migrate towards the vasculature. On the other hand, the existence of two MK pools (fast vs. more quiescent response) might explain why therapies to increase platelet counts are not fully effective. We believe that further studies addressing exactly this concept might ultimately provide insights on how to activate the second quiescent MK pool to accelerate platelet production.

## Methods

**Mice.** All animal experiments were approved by the district government of Lower Frankonia (Bezirksregierung Unterfranken). For all experiments, 8–12 week old male and female mice (equal distribution in both groups) were used.

C57BL/6JRj mice were obtained from Janvier Labs. Mice lacking talin1 specifically in MKs and platelets (Tln$^{fl/fl, PF4-Cre}$; on a C57BL/6J background) have been described previously[32]; Cre-negative litter mates served as control animals. For the EdU treatment, each mouse (~30 g) was daily fed with 25 mg EdU embedded in glucose-agarose cubes for 10 days prior to platelet depletion and subsequently until the mice were euthanized. These experiments were performed according to institutional guidelines by the district government of Berlin, Germany.

**Antibodies and reagents.** Anesthetic drugs (medetomidine (Pfizer, Karlsruhe, Germany), midazolam (Roche, Grenzach-Wyhlen, Germany) and fentanyl (Janssen—Cilag, Neuss, Germany)) were used according to the local regulations. Mice were bled in high-molecular-weight heparin (Ratiopharm, Ulm, Germany). Mice were further treated with Plerixafor (AMD3100, Selleckchem, Munich, Germany; 5 mg/kg bodyweight) or SRP4388 (SDF1 alpha, Sigma-Aldrich, Schnelldorf, Germany; 16 µg/kg bodyweight). Antibodies were fluorescently labeled using protein labeling kits (Alexa Fluor 488/546/594/647/750, Life Technologies, Darmstadt, Germany). Anti-CD105 antibody (BioLegend, San Diego, CA, USA) or tetramethylrhodamine dextran (2 MDa; Life Technologies, Darmstadt, Germany) were used for BM vasculature staining. All other antibodies were generated and modified in our laboratory as previously described[19]

**Platelet depletion.** Thrombocytopenia was induced in C57BL/6JRj WT mice by intravenous injection of rat anti- GPIbα (CD42b; Emfret Analytics, Eibelstadt, Germany; 2.0 µg/g body weight). Peripheral platelet counts were determined using a fully automated hematology analyzer and additionally by flow cytometry.

**Cryo-sections.** Femora and sterna of male and female mice age 8–12 weeks were isolated, fixed with 4% PFA (AppliChem GmbH, Darmstadt, Germany) (AppliChem) in 5 mM sucrose in phosphate buffered saline (PBS), transferred into 10% sucrose in PBS and dehydrated using a graded sucrose series (20%, 30%). Subsequently, the samples were embedded in SCEM medium (Section Lab, Hiroshima, Japan), snap-frozen in liquid nitrogen and stored at −80 °C. 10 µm-thick cryosections were generated according to the Kawamoto protocol[13] at a Leica CM1900 cryotom (Leica, Mannheim, Germany). For EdU-stainings, sections were rehydrated 20 min with PBS followed by permeabilization 30 min with 0.5% Triton X-100. After three 5 min washes with washing buffer (PBS; 0.1% Tween 20; 5% FCS), the click-it reaction was performed for 30 min according to the manufacturer's instructions (Click-iT® EdU Alexa Fluor® 647 Imaging Kit, Thermo Fisher). MKs were stained with anti-CD41-FITC antibody (f.c. 5 µg/ml; Becton Dickinson, Franklin Lakes, USA), anti-β1-tubulin antibody (f.c. 10 µg/ml; Sigma-Aldrich, Schnelldorf, Germany), or anti-GPIX (f.c. 5 µg/ml; CD42a)

antibody. Secondary antibodies were purchased as conjugates with Alexa Fluor 488/594/647 (f.c. 7 µg/ml; Life Technologies, Darmstadt, Germany). Nuclei were stained with Hoechst33258 (Biomol, Hamburg, Germany) and mounted with Mowiol (Sigma-Aldrich) or 4′,6-diamidine-2′-phenylindole dihydrochloride (DAPI)-containing mounting medium (DAPI Fluoromount-G; Southern Biotech, Birmingham, AL, USA). Staining with antibodies and secondary reagents was performed for 45 min at room temperature (RT). Sections were analyzed by confocal microscopy (Leica TCS-SP5) and processed with the Leica LAS AF software and ImageJ software (NIH). MK number and density were measured in every visual field. Three femur sections from three separate animals were examined.

**Megakaryocyte adhesion and proplatelet formation.** Fetal liver cells of wildtype mice were isolated from E14.5 old embryos and cultivated for 3 days at 37 °C and 5% CO$_2$. Three hours prior to the enrichment of MKs via a bovine serum albumin (BSA) gradient, the cells were incubated with IgG control antibodies (Emfret Analytics, Eibelstadt, Germany), anti-α2-integrin (LEN/B[49]) or anti-GPIX derivatives at 10 µg/mL or remained untreated. To determine proplatelet formation, 17,000 MKs/cm$^2$ were seeded in triplicates and incubated at 37 °C and 5% CO$_2$ for 24 h. For the adhesion assay 140,000 MKs were seeded on BSA or human fibrillar collagen I (Corning; 10 µg/cm$^2$) coated coverslips (12 mm diameter) and incubated for 4 h at 37 °C and 5% CO$_2$. Afterwards the non-adherent cells were removed and the adhered MKs were stained with an FITC conjugated anti-CD41 antibody (f.c. 5 µg/ml; Becton Dickinson, Franklin Lakes, USA) for 1 h before mounting them with DAPI. At least 10 visual fields per cover slip were recorded at a TCS SP8 (Leica) confocal microscope and the surface coverage analyzed with FIJI[50].

**Multiphoton intravital microscopy.** Mice were anesthetized by intraperitoneal injection of medetomidine 0.5 mg/g, midazolam 5 mg/g and fentanyl 0.05 mg/g body weight. A 1-cm incision was made along the midline to expose the frontoparietal skull, while carefully avoiding damage to the bone tissue. The mouse was placed on a custom metal stage equipped with a stereotactic holder to immobilize the head. BM vasculature was visualized by injection of tetra-methylrhodamine dextran (8 µg/g body weight) and anti-CD105 Alexa Fluor 546/Alexa Fluor 594 (0.4 µg/g body weight). Platelets and MKs were visualized by injection of Alexa Fluor 488-labeled anti-GPIX derivative (0.6 µg/g body weight). Images were acquired at a frame rate of 12 f/min or 1 f/min on an upright two-photon fluorescence microscope (TriM Scope I multiphoton system, LaVision BioTec, Bielefeld, Germany or TCS SP8 MP, Leica Microsystems, Wetzlar, Germany) equipped with a ×20 water objective with a numerical aperture of 0.95. Emission was detected with HQ535/50-nm and ET605/70-nm filters. A tunable broad-band Ti:Sa laser (Chameleon, Coherent, Dieburg, Germany) was used at 760 nm to capture Alexa Fluor488 and Alexa Fluor 546/Alexa Fluor 594/tetramethylrhodamine dextran fluorescence. FIJI[50] and Imaris® 8.3.1 (Bitplane, Zürich, Switzerland) were used to generate movies.

**Tissue preparation for LSFM.** Platelets/MKs and BM vasculature were stained by injecting Alexa Fluor 750-labeled anti-GPIX derivative (0.6 µg/g body weight) and anti-CD105 Alexa Fluor 647 (0.4 µg/g body weight), respectively. Thirty minutes after injection the mice were transcardially perfused with ice-cold PBS to wash out the blood and ice-cold 4% paraformaldehyde (PFA, Sigma-Aldrich, Schnelldorf, Germany (pH 7.2)) to fix the tissues. Sterna, skull, and femora were harvested and stored in 4% PFA for 30 min. Bone samples were washed in PBS and decalcified in 10% ethylenediaminetetraacetic acid (AppliChem, Darmstadt, Germany) for 96 h at 4 °C on a shaker. Samples were then washed in PBS, followed by dehydration in a graded methanol (Sigma-Aldrich) series (50%, 70%, 95%, 100% for 30 min each) at RT and stored at 4 °C overnight. The methanol was replaced stepwise by a clearing solution consisting of one part benzyl alcohol to two parts benzyl benzoate (BABB, Sigma-Aldrich). After incubation in the clearing solution for at least 2 h at RT, tissue specimens became optically transparent and were used for LSFM imaging on the following day.

**LSFM setup and data acquisition.** Image acquisition of cleared samples was performed in a home-built scanning light-sheet fluorescence microscope similar to previous designs[18]. The specimen was affixed with cyanoacrylate glue (Weicon, Münster, Germany) to a glass micropipette (100 µl, Brand GmbH) that was mounted on a motorized (8CMA06-25/15 Standa, Vilnius, Lithuania) four axis stage (xyz translation (Newport, Darmstadt, Germany) and rotation (Standa). A home-built coverglass chamber filled with BABB solution allowed imaging of the cleared sample immersed in clearing solution. A custom fiber coupled laser combiner (BFI OPTiLAS GmbH, Groebenzell, Germany) with two red laser lines (642 and 730 nm) was used for fluorescence excitation. For collimation, an objective lens (A10/0.25 Hund, Wetzlar, Germany) providing a beam diameter of roughly 3 mm was used. All laser lines were joined via a dichroic beamsplitter (DCLP 660, AHF Analysentechnik, Tübingen, Germany). A galvanometric scanner (6210 H, Cambridge Technology, Bedford, MA, USA) elongated the resulting laser beam with a frequency of 600 Hz before being focused via a theta lens (VISIR f. TCS-MR II, Leica, Mannheim, Germany) to create a virtual light sheet. The focused beam was relayed with a tube lens and an objective lens (EC Plan-Neofluar 5x/0.16 M27, Zeiss, Göttingen, Germany) to the sample.

For detection, a HCX APO L20x/0.95 IMM objective (Leica, Wetzlar, Germany) was mounted on a translation stage (Newport) perpendicular to the light sheet. The fluorescence emission was spectrally filtered using a motorized filter wheel (MAC 6000 Filter Wheel Emission TV 60 C 1.0x (D)) with a MAC 6000 Controller (Zeiss, Göttingen, Germany) equipped with the following filters according to the fluorescence of the used fluorescent labels: 491 nm BrightLine HC 525/50 (autofluorescence), 642 nm HQ 697/58 (Alexa Fluor 647), and 730 nm BrightLine HC 785/62 (Alexa Fluor 750) (AHF). The image was generated by an infinity-corrected ×1.3 tube lens ∞/240–340 (098.9001.000, Leica, Wetzlar, Germany) and detected by a sCMOS camera Neo 5.5 (2560 × 2160 pixels, 16.6 mm × 14.0 mm sensor size, 6.5 µm pixel size, Andor, Belfast, UK). Synchronization of the laser illumination, image acquisition, focus correction, and z scanning was controlled by IQ 2.9 software (Andor). Multicolor stacks were generated by imaging the three color channels sequentially in each plane in increments of 2 µm. Images were saved as Tagged Image File Format (TIFF) stacks.

**Image analysis**. Multicolor LSFM stacks were processed and analyzed by FIJI[50] and Imaris® 7.7.2 and 8.3.1 (Bitplane AG, Zurich, Switzerland). Data visualization and analysis was performed in three major steps: (i) image preprocessing, (ii) segmentation, (iii) data extraction. Step (i) was performed using either Imaris or FIJI, steps (ii) and (iii) were performed using Imaris® 7.7.2 and 8.3.1 including the ImarisCell module. Statistical analysis of object positions, numbers, and volumes (MK, vessel, BM), edge-to-edge distance from MK to vessel or bone, and available vessel interspace were performed using Microsoft Excel (ver. 2013, Microsoft Corporation, Redmond, WA, USA), Origin (ver. 8.6, OriginLab Corporation, Northampton, MA, USA), and SPSS Statistics (ver.23, IBM, Ehningen, Germany) and Matlab software (Mathworks, USA). MKs were considered to be vessel-associated MKs if the maximal distance between the MK edge and the vessel was one voxel, where a voxel has the dimensions $0.5 \times 0.5 \times 2$ µm$^3$. All other MKs are referred as non-vessel-associated. Bone structures were derived from autofluorescence by using the pixel classification algorithm of the well-established interactive learning and segmentation toolkit ilastik[51]. Bone structures derived from ilastik were exported to Imaris 8.3.1 for segmentation and further analysis.

Multi-photon microscopy images were corrected for noise and converted to binary images using FIJI. MKs that did release proplatelets during the observation period were manually excluded prior to the cell tracking analysis. Cell tracking of binary images was performed by Imaris 8.3.1 or with the active contours module of Icy software (www.icy.bioimageanalysis.org, Institut Pasteur, France), where the software calculated the center of mass of the cells from the binary images for each frame. Mean square displacement analysis (MSD) of cell trajectories was performed by using Matlab (Mathworks) and msdanalyzer class[52]. According to the diffusion theory for two-dimensional data, the mean squared displacement is calculated for different time scales and linked to these by

$$\mathrm{MSD} = 4K_\alpha t^\alpha \qquad (1)$$

where $t$ is the time scale, $K_\alpha$ the time scale dependent diffusion coefficient, and $\alpha$ the exponential coefficient of movement. While $K_\alpha$ lacks a distinct physical meaning, exponent $\alpha$ defines different motion types as:

$\alpha = 1$, for Brownian diffusion

$\alpha < 1$, for subdiffusion or hindered diffusion

$\alpha > 1$, for superdiffusion or enhanced diffusion

**Computer simulations**. Simulations were performed with a self-written algorithm in Matlab software (Mathworks) using volume image stacks of mouse sterna acquired by LSFM and processed with Bitplane Imaris® 7.7.2 software (see above). For random simulation (SimR), MKs were randomly placed in the 3D BM space. For NVA random simulation (SimNVA), BM vasculature and vessel associated MKs served as space-limiting scaffold. The remaining MKs were randomly placed into the intervascular region without vessel or cell contact. The MK-vessel distances were calculated using vessel distance transformation maps. For both types of simulations, the number and size distribution of MKs in the pool were matched to the experimentally estimated values. The presented results are mean ± SD from at least three independent experiments per group, if not stated otherwise.

**Statistical analysis**. Differences between two groups were statistically analyzed using the Mann–Whitney $U$ test and difference between three or more groups were analyzed using one-way ANOVA. $P$-values > 0.05 were considered as not significant ($p > 0.05$). $P$-values < 0.05 were considered statistically significant with: *$p < 0.05$, **$p < 0.01$, ***$p < 0.001$.

**Data availability**. The datasets generated and/or analyzed in the current study, as well as the source code of the computational simulations are available from the corresponding authors on reasonable request.

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

## Acknowledgements

This work was supported by the Deutsche Forschungsgemeinschaft (SFB688) and the Rudolf Virchow Center. J.M.M.v.E. was supported by a grant of the German Excellence Initiative to the GSLS, University of Würzburg. We thank Anja Hauser (DRFZ, Berlin) for help with the EdU experiments, Paquita and Alan Nurden for fruitful discussions, Ann Wehman for proofreading the manuscript and Stefanie Hartmann for excellent technical assistance.

## Author contributions

Conceptualization, D.S., H.S., B.N., and K.G.H.; Methodology, J.M.M.v.E., O.A., M.G., J.P., P.S., M.F., C.B., A.B., and K.G.H.; Software, O.A., M.G., P.S., and K.G.H.; Formal Analysis, J.M.M.vE., O.A., M.G., and K.G.H.; Investigation, D.S., J.M.M.v.E., O.A., D.Se., and S.D.; Writing–Original Draft, D.S.; Writing–Review and Editing, J.M.M.v.E., H.S., B.N., K.G.H.; Visualization, J.M.M.v.E., O.A.; Funding Acquisition, D.S., B.N., K.G.H.; Supervision, D.S., H.S., B.N., and K.G.H.

## Additional information

**Competing interests:** The authors declare no competing financial interests.

