## [Peer Review File · Nature Communications]

Reviewers' comments:

Reviewer #1 (Remarks to the Author):

Stegner et al describe some very elegant imaging work unravelling the dynamics of megakaryocytes in the bone marrow. They use a combination of histological sections, light sheet microscopy and intravital microscopy to achieve detailed observations of the morphology, distribution and potential migration of megakaryocytes as they mature and reach the conclusion that these very large cells do not move across the bone marrow space. Moreover, they use systematic analysis of distance measurements to evaluate the distribution of megakaryocytes relative to vessels and across the bone marrow space, using the ideal control of randomly positioned artificially generated cells.

My main concern is about the conclusions reached: with the data presented, it is hard to infer any specific regulatory mechanism of vasculature over megakaryocytes. Rather, it seems that vasculature affects/dictates the distribution of most megakaryocytes. Because of this, it is important that the authors are very careful about defining what they recognise as homogeneous versus not. My understanding is that only the non-vessel associated megakaryocytes are homogeneously distributed across the bone marrow space.

Specific points:

Figure 1. 30 minutes time lapse is very short to determine migration of slow-moving cells. The authors present a supplementary movie acquired for a much longer time frame, which is more convincing. In panel B, the tracks actually seem inconsistent with the conclusion that the cells do not migrate. How were they generated?

Figure 1, 4 and supplementary movie 1. CD105 is not the most widely used marker of vasculature. It seems to work well in sections, but less so in whole mounts and in vivo images. The authors mention they used also CD31 and dextran. Could some of those images be used instead? Panel G seems to show some different dynamics between vessels associated and non associated megakaryocytes, rather than a homogeneous response. Also, because the vessel-associated MKs most often have a higher proportion of EdU positive or bright cells, are the authors certain that EdU has reached non vessel-associated MKs as efficiently? It could be useful to show dose-response data.

I don't understand how the fact that the EdU stained proportion of MKs following platelet depletion is initially higher but later returns to normal is an indication that MKs are distributed homogeneously.

Minor points:

In figure 4G, the EdU accumulation is a reflection of new DNA synthesis rather than a measure of DNA synthesis rate. The y axis label of 'phase abundancy' is very counter intuitive to interpret. '% of total MK in the category' may better reflect what the authors mean; alternatively, the data could be grouped differently.

The text refers to a figure 4H which does not exist.

Reviewer #2 (Remarks to the Author):

In this manuscript, Stegner and colleagues provide new insights into the spatial regulation of megakaryocytes in the bone marrow space. In this top-notch manuscript, they carefully analyze the distribution of megakaryocytes within the bone marrow using a two-photon intravital approach and come up with a new model for platelet production. They very elegantly show that megakaryocytes are

predominantly sessile cells in close contact with the vasculature and homogeneously distributed in the marrow. They also demonstrate that induction of thrombocytopenia does not alter motility/localization and clearly show that inhibition of the CXCR pathway or talin depletion does not influence megakaryocyte distribution. Finally, they use a computational model that supports a vessel-biased distribution. Overall, the work is highly significant because it challenges the current model of megakaryopoiesis, in which blood cell precursors migrate from an endosteal niche towards the vessel sinusoids during maturation. I think that this is a very solid manuscript that is highly appropriate for Nature Communications, and the work will have a major and sustained impact in the field. I do have some concerns/suggestions that should strengthen the manuscript.

1. The authors should consider the possibility that the fluorescent antibody they are using to label live megakaryocytes could influence platelet production. Were any control experiments done to address this possibility?

2. The authors clearly show that megakaryocytes do not leave the marrow and enter the circulation and lung to make platelets. They should highlight, expand, and emphasize this point. They should also emphasize that they do not observe megakaryocyte rupture.

3. The authors should consider looking more closely at MK progenitors and early/immature MKs. The authors should consider sorting the cells by ploidy and then distance to the vessels. I think they need to look at more progenitor-like cells if they are going to propose the model in Fig 6 and make statements like 'MKs ... are replenished by progenitors in close proximity to the vasculature.' They do a very convincing job of showing that mature MKs are near vessels, but still haven't convinced me that the progenitors don't migrate to the vessel.

4. The authors state that platelet recovery after depletion is due to increased MK turnover. However, they don't measure that directly. They only see increased endomitosis - which suggests increased/faster MK maturation. I think that this is an important point because there needs to be a good explanation for the platelet replenishment in order for this to have physiological relevance. Perhaps this could be addressed with a classical pulse-chase where you look how many labeled MKs are left after certain time points. This would directly address cell turnover and not nuclear content accumulation.

5. What are the platelet counts in the CXCR and Talin deficiency models? If there is a decrease in platelet count, what accounts for this change since the MK number, distribution, and volumes are unchanged?

6. Much of the authors' work (including all of the migration data, I believe) was done using CD42a as a marker, which is a marker of mature(ish) MKs. The authors should be careful not to inappropriately extrapolate their observations made on CD42a+ cells to all MK lineage cells, as the MK progenitors are presumably CD42a-. The simultaneous use of CD34 or another similar marker would be useful to positively identify MK progenitors.

7. The authors are making an inappropriate presumption that the endosteal niche is devoid of blood vessels. That is old dogma that has long been refuted (PMC4514480, PMC4156024, PMC3984353, PMC4148778 (commenting on PMC3821873), PMC4850557). Ideally, in addition to the distance to the nearest vessel, the distance to the bone should be calculated too.

8. The authors demonstrated quite convincingly that mature(ish) MK's (i.e. CD42a+) have low motility and were distributed around the bone marrow, localized near vessels. However, this is not necessarily in such stark contrast with the model proposed by Rafii. The model (see Fig 6C in his Jan 2004 Nat

Med paper) is that HSPCs at the osteoblastic/endosteal niche respond to TPO and generate MK progenitors which then migrate towards the sinusoidal vessels where they undergo maturation, extend proplatelets, and produce platelets into circulation. In my opinion, there is not sufficient evidence in the current paper to completely invalidate this model, since MK progenitor migration was not assessed, and distance to bone was never measured. That said, recent data, e.g. PMC4850557, has indicated that most HSCs, including quiescent HSCs, are distributed throughout the bone marrow and not near bone, so the current authors' hypothesis is certainly plausible, even likely. This should be addressed.

9. What is the relevance to human platelet production in both health and disease? This should be highlighted and clearly, concisely stated in the discussion and abstract.

Reviewer #3 (Remarks to the Author):

In this paper Stegner et al. used live two-photon microscopy imaging, computational simulations and immunofluorescence of mouse bone marrow to demonstrate that megakaryocytes do not migrate from periostic to sinusoidal niche but rather accommodate in close contact with sinusoids. Further, platelet depletion, CXCR4 inhibition or talin deficiency seems not to affect megakaryocyte motility or localization in vivo. Their conclusion is that MKs originate and are replenished by sinusoid-resident precursors. However, the concept that MKs are mainly sinusoid-associated is not surprising nor completely original as recently published by the same group (Semeniak et al. J Cell Sci. 2016 Sep 15;129(18):3473-84).

Data on MK motility, even though in agreement with previous work by Junt et al. (Science. 2007 Sep 21;317(5845):1767-70), would be of interest however some concerns temper enthusiasm:

- Why do the authors expect a high rate of motility in GpIX+ mature MKs, which are already in close contact with sinusoids? Especially in a time window of 30 minutes? Differently from CD41, GpIX is considered a late maturation marker in MKs. Is it possible that immature MKs were excluded from the live analysis within the proposed experimental settings? CD41/ β 1 Tub Double staining in Fig 2C shows the existence of two MK pools. How is GpIX expression distributed among these MK populations?
- MK migration in Figure 1 was measured in 37 cells by 2P-IVM. Which cells were selected for these measurements? Were they exclusively vessel-associated MKs or randomly distributed cells? Were MKs associated with periostic osteoblasts included in the analysis? Why wasn't a comparison carried out between periostic-associated MKs vs sinusoids-associated MKs in terms of instantaneous migration? In order to exclude a potential MK origin from the endosteal niche, this site should be analyzed as well. Periostic niche is present only in the immunofluorescence of Fig. 2 of sternum and femora. Is cranial BM suitable for analyzing endosteal-MK contacts to a similar extent of BM femur/sternum? This topic is only discussed briefly in the discussion section.

In general, the conclusions of this manuscript are speculative with respect to the data presented.

Major Concerns:

- A major point of concern is the use of an anti-GpIX antibody as the live tracking system for MK visualization. As stated by authors, this labeling scheme was required to overcome the limitations of the CD41-YFP mouse published by Junt et al. 2007 in Science. However, details of the source of the GpIX antibody are not provided (e.g., clone, epitope, manufacturer) and several missing controls compromise the major premises of this paper. For example, experiments demonstrating negligible effects of this GpIX antibody on MK function (PPF, Adhesion/migration) in vitro should be provided. What are the effects of GpIX ab injection on mice peripheral blood parameters?

- Why weren't experiments in mice with CXCR4 inhibition or talin deficiency performed by 2P-IVM? In this regard the authors should explain why MK distribution in mice with CXCR4 inhibition or talin deficiency was assessed by immunofluorescence with an anti-CD41 antibody that recognizes both immature and mature cells (and not the same GpIX ab)?

- What concentration of SDF-1 was used? Was the protocol different from that of Niswander et al. Blood. 2014 Jul 10;124(2):277-86? The authors should explain or at least discuss discrepancies with this work. Was the peripheral blood platelet count affected by AMD3100 or SDF-1 treatment?
- Description of results are too speculative and conclusions not fully supported by experiments. For example, the increased rate of cell ploidy after platelet depletion is described as "reflecting the increased MK turnover due to elevated platelet demand" but this finding was not experimentally tested. "MK turnover" or "MK replenishment by precursors" are terms found within the manuscript but these concerns were not experimentally assessed.

Minor Comments:

- Figure 6A is not clear.
- Is there a reason for injecting a 1000-fold more concentration of GpIX ab with respect to previously published paper by the same group (0.6µg/g mice in Bender et al. Nat Commun. 2014 Sep 4;5:4746. doi:10.1038/ncomms7507 vs 0.6mg/g mice in this paper)?

Reviewers' comments:

Reviewer #1 (Remarks to the Author):

Stegner et al describe some very elegant imaging work unravelling the dynamics of megakaryocytes in the bone marrow. They use a combination of histological sections, light sheet microscopy and intravital microscopy to achieve detailed observations of the morphology, distribution and potential migration of megakaryocytes as they mature and reach the conclusion that these very large cells do not move across the bone marrow space. Moreover, they use systematic analysis of distance measurements to evaluate the distribution of megakaryocytes relative to vessels and across the bone marrow space, using the ideal control of randomly positioned artificially generated cells.

My main concern is about the conclusions reached: with the data presented, it is hard to infer any specific regulatory mechanism of vasculature over megakaryocytes. Rather, it seems that vasculature affects/dictates the distribution of most megakaryocytes. Because of this, it is important that the authors are very careful about defining what they recognise as homogeneous versus not. My understanding is that only the non-vessel associated megakaryocytes are homogeneously distributed across the bone marrow space.

We thank the Reviewer for the positive feedback. The Reviewer stated “the vasculature seems to dictate the distribution of most megakaryocytes” and this phrase describes precisely the situation within the bone marrow. Thus, we decided to follow up on it to strengthen the discussion part and, most importantly, avoid misunderstandings. In detail, we have now removed the word ‘homogenously’ to avoid any confusion. Instead, we highlighted particularly

- (1) **the absence of MK-deficient regions** in the bone marrow
- (2) **the limited space** for MKs within the bone marrow vasculature which
- (3) **forces the majority of MKs to be in direct contact with the vessels.**
- (4) **the high similarity of bone-associated and bone-distant bone marrow**
- (5) the random distribution of **non-vessel associated MKs** within the bone marrow cavity.

We are convinced that our changes increased the clarity of our manuscript.

Specific points:

Figure 1. 30 minutes time lapse is very short to determine migration of slow-moving cells. The authors present a supplementary movie acquired for a much longer time frame, which is more convincing. In panel B, the tracks actually seem inconsistent with the conclusion that the cells do not migrate. How were they generated?

The traces depicted in Figure 1B of the initial submission of our manuscript depict the movement of our MKs within this 30 min time frame. The apparent movement does, however, reflect rather a ‘wobbling’ of MKs than a migration from one position to another. MKs in contact with the vessel and releasing pro-platelets during the observation period have been excluded, as the formation of proplatelets into the vessel lumen would have resulted in an apparent movement of the center of mass as well, even though the MK-body did not alter its position.

For the revised version of our manuscript, we have generated approximately **3 hours of intravital videos** per mouse and analyzed **six mice**, reflecting in total 54 MKs which did not make proplatelets. Eight of these MKs were not in contact of the vasculature. These data are now included in the modified Figure 1 of the revised manuscript and in movie 1. However, the MSD analysis and the MK traces are similar to those of the shorter videos, as we could not observe a significant movement of MKs within the 3 h observation period. Thus, our initial conclusions that MKs are sessile and do not migrate *in vivo* is confirmed by the longer recordings.

Figure 1, 4 and supplementary movie 1. CD105 is not the most widely used marker of vasculature. It seems to work well in sections, but less so in whole mounts and *in vivo* images. The authors mention they used also CD31 and dextran. Could some of those images be used instead?

We do agree with the Reviewer that CD31 (PECAM-1) is a more frequently used endothelial marker than CD105. However, PECAM-1 is additionally expressed on MKs and platelets and has been shown to be a negative regulator of laminin-induced platelet activation (Crockett *et al.*, JTH 2010; 8:1584-93). Moreover, it has been proposed to influence platelet release (Dhanjal *et al.*, Blood 2007; 10:4237-44) therefore, we decided for our study against the use of anti-CD31 antibodies for any *in vivo* labeling. MKs would inevitably be stained with anti-CD31 antibodies (see Figure to Reviewers below), which would make it very difficult to distinguish between MKs/proplatelets within the vessel lumen and those still residing in the bone marrow. As we could see in bone marrow sections identical stainings of the vasculature for anti-CD31 and anti-CD105 antibodies, but only very faint labeling of MKs with anti-CD105 antibodies (see Figure 1 to Reviewers below) we consider the anti-CD105 staining a valid marker for the endothelial lining. Moreover, for all of our intravital videos the vasculature has always been stained with anti-CD105 antibodies (labeling the lining of the vessels) in addition to a high-molecular fluorophore-labeled dextran to stain for the vessel lumen providing a clear vasculature staining.

Figure to Reviewers 1. Megakaryocytes express CD31. A) Femur cryosection stained with anti-CD31 (red), anti-CD105 (blue), anti-CD41 (green) and counterstained with DAPI (grey) demonstrates that megakaryocytes are positive for CD41 and CD31, while the endothelial lining is stained by anti-CD31 and anti-CD105 antibodies. Scale bar 100 μ m. B) Two-photon intravital microscopy (2P-IVM) reveals that the anti-CD31 antibody not only stains the endothelial lining, but in addition, it does stain MKs (stained with anti-CD42a antibodies, green). Scale bar 30 μ m.

Panel G seems to show some different dynamics between vessels associated and non associated megakaryocytes, rather than a homogeneous response. Also, because the vessel-associated MKs most often have a higher proportion of EdU positive or bright cells, are the authors certain that EdU has reached non vessel-associated MKs as efficiently? It could be useful to show dose-response data.

We do understand the Reviewer's concern and thus would like to stress two important points:

1) We have observed EdU positive cells, other than MKs, within the bone marrow (see Figure to Reviewers 2 below), while some vessel associated cells were EdU negative or only weakly stained. Thus, we are convinced that the different EdU intensity is not due to different accessibility of EdU depending on the distance to the vessel.

2) The EdU dose-response curve has a steep slope (since a certain level of EdU is required to be bioavailable for incorporation into the DNA) and quickly reaches a plateau (Zeng *et al.*, Brain Res 2010;

1319C:21-32), resulting in a narrow window for dose-response related differences. For this reason, we could not get an animal experimentation license for this experiment and cannot provide further data on this issue. However, as we can rule out that EdU-distribution was limited to the vessel-associated BM we are confident that EdU distribution was no limiting factor of our analyses.

Figure to Reviewers 2. Early EdU-staining is not restricted to the proximity of the vasculature. Femurs were sectioned and stained for EdU (blue), which is incorporated into nuclei (grey). MKs are stained with anti-CD41 antibodies (green) and vessels with anti-CD105 antibodies (red). EdU incorporation was detected in vessel-associated and non-vessel-associated cells. Scale bar 50 μ m.

I don't understand how the fact that the EdU stained proportion of MKs following platelet depletion is initially higher but later returns to normal is an indication that MKs are distributed homogeneously.

We apologize for apparently not being sufficiently clear on this point. The fact that the EdU-stainings differ only transiently between the two groups demonstrates "robustness of the MK distribution and thrombopoiesis even under conditions of high platelet production" as we stated. We meant that the increased platelet demand results in accelerated platelet production and MK responses concerning one subset of MKs which is reflected by the higher proportion of EdU-positive MKs at day 3 after platelet depletion. However, at later time points the system 'bounces back' to 'normal' as the fraction of EdU-negative (= non-DNA replicating MKs) is similar between platelet-depleted and unchallenged mice on day 5 and 7. Likewise, the fraction of strongly stained MK fraction is comparable between the two groups. **In conclusion, there seems to be a 'fast-response' MK (and presumably MK progenitor) fraction and a more quiescent MK pool.** With regard to the localization we can only state that the vessel-associated MKs have in general a higher proportion of EdU-positive cells, indicating that the 'fast-responders' are already at the sinusoids (which does make sense from a biological point-of-view). However, we cannot conclude from our EdU stainings whether the MKs are homogeneously distributed or not. We have adapted our discussion section (p. 16), to avoid this misunderstanding.

Minor points:

In figure 4G, the EdU accumulation is a reflection of new DNA synthesis rather than a measure of DNA synthesis rate. The y axis label of 'phase abundancy' is very counter intuitive to interpret. '% of total MK in the category' may better reflect what the authors mean; alternatively, the data could be grouped differently.

We appreciate the Reviewer's comment and have changed our labeling accordingly.

The text refers to a figure 4H which does not exist.

We apologize for our mistake which has now been corrected by deleting this reference as Figure 4H had been inserted in Figure 4G (now Figure 5G) before the initial submission.

Reviewer #2 (Remarks to the Author):

In this manuscript, Stegner and colleagues provide new insights into the spatial regulation of megakaryocytes in the bone marrow space. In this top-notch manuscript, they carefully analyze the distribution of megakaryocytes within the bone marrow using a two-photon intravital approach and come up with a new model for platelet production. They very elegantly show that megakaryocytes are predominantly sessile cells in close contact with the vasculature and homogeneously distributed in the marrow. They also demonstrate that induction of thrombocytopenia does not alter motility/localization and clearly show that inhibition of the CXCR pathway or talin depletion does not influence megakaryocyte distribution. Finally, they use a computational model that supports a vessel-biased distribution. Overall, the work is highly significant because it challenges the current model of megakaryopoiesis, in which blood cell precursors migrate from an endosteal niche towards the vessel sinusoids during maturation. I think that this is a very solid manuscript that is highly appropriate for Nature Communications, and the work will have a major and sustained impact in the field. I do have some concerns/suggestions that should strengthen the manuscript.

We appreciate that the Reviewer considers our study to be solid and of relevance for Nature Communications. We provide here additional data and are confident that we have addressed the raised concerns.

1. The authors should consider the possibility that the fluorescent antibody they are using to label live megakaryocytes could influence platelet production. Were any control experiments done to address this possibility?

The Reviewer raises an important point. We have indeed performed a series of experiments to determine the effect of the used antibodies on platelet formation. First, we have assessed the impact of our anti-GPIX derivatives on proplatelet formation in an *in vitro* assay and have tested whether the presence of the antibody would affect MK surface coverage on collagen I. However, proplatelet-formation was unaffected, as was the surface coverage of MKs on collagen I (see new Supplementary Figure 1). In contrast, the integrin $\alpha 2\beta 1$ -blocking antibody LEN/B resulted in reduced surface coverage on collagen I, in line with our previous study (Semeniak *et al.*, J Cell Sci 2016; 129:3473-84).

To assess potential effects on platelet production *in vivo*, we determined platelet counts in vehicle treated mice and mice receiving 1.5 $\mu\text{g/g}$ bodyweight (more than twice the doses used for multiphoton or LSM microscopy) of the fluorophore-coupled anti-GPIX derivative. However, we could not observe differences between the two groups, confirming the *in vitro* data that our anti-GPIX derivative does not affect platelet formation (see Figure to Reviewers 3). Of note, anti-GPIX derivatives are widely used in the field to label platelets for *in vivo* thrombosis models (e.g. May *et al.*, Blood 2009; 114:3464-72, Deppermann *et al.*, J Clin Invest 2013; 123:3331-42) or for the determination of platelet life span (e.g. Grosse *et al.*, J Clin Invest 2007; 117:3540-50, Deng *et al.*, Nat Comm 2016; 7:12863). Moreover, we have previously used this antibody to stain MKs for intravital microscopy (Bender *et al.*, Nat Commun 2014; 5:4746). In conclusion, we are convinced that our labeling strategy does not affect platelet production. We thank the Reviewer for raising this important point and have included this information into the text of our revised manuscript (see page 5).

Figure to Reviewers 3. The anti-GPIX antibody (p0p6) derivative, which has been used for 2P-IVM does not affect platelet counts. Male C57Bl/6J mice received either 1.5 μ g Alexa488-conjugated anti-GPIX derivative/g body weight (p0p6 derivative, grey bars) or vehicle (black bars) and platelet counts were determined by flow cytometry. Depicted are mean \pm standard deviation of relative platelet counts (compared to day 0).

2. The authors clearly show that megakaryocytes do not leave the marrow and enter the circulation and lung to make platelets. They should highlight, expand, and emphasize this point. They should also emphasize that they do not observe megakaryocyte rupture.

We appreciate the Reviewer's valuable suggestion, particularly in light of the recent publication on platelet biogenesis in the lung (Lefrancais *et al.* Nature 2017; 544:105-9). Indeed, we barely find MKs within the BM vasculature using LSFM microscopy ($2.0 \pm 1.6\%$ of all MKs) and did not observe transmigration of whole MKs in our 2P-IVM (we only found 2 out of 230 MKs within the BM vasculature), making it unlikely that MK migration from the BM to the lung is as important as proposed by Lefrancais and colleagues. We have implemented this into the discussion (p.15) of our manuscript.

Besides 'classical' proplatelet formation, MK rupture has been suggested as a second way of platelet release (Nishimura *et al.*, J Cell Biol 2015; 209:453-66). As suggested by this Reviewer, we now mention in the revised text on page 14 that we did not observe MK rupture in our experimental setting. However, Nishimura and colleagues have proposed that MK rupture is relevant particularly under inflammatory conditions, an experimental setting which we did not analyze in our study.

3. The authors should consider looking more closely at MK progenitors and early/immature MKs. The authors should consider sorting the cells by ploidy and then distance to the vessels. I think they need to look at more progenitor-like cells if they are going to propose the model in Fig 6 and make statements like 'MKs ... are replenished by progenitors in close proximity to the vasculature.' They do a very convincing job of showing that mature MKs are near vessels, but still haven't convinced me that the progenitors don't migrate to the vessel.

The Reviewer's point is well taken. The reason why we did not include studies on progenitor cells into our manuscript is the lack of a suitable surface marker that is specific for early megakaryocytic progenitor cells. While it would be possible to isolate these cells for ploidy analysis, we would obviously lose all structural information regarding their distance to vessels. CD41 has been proposed to be expressed on megakaryocyte-erythroid progenitors (MEPs), which do not yet express the GPIb-V-IX complex (Pang *et al.*, J Clin Invest 2005; 115:3332-8). While this is mostly true for human MKs, in mice,

we found virtually no CD41+ MKs that were negative for CD42 (see new Supplementary Figure 1). Therefore, a simple co-staining approach was not sufficient to solve the issue described. We have tested several antibody combinations to specifically stain MK progenitors. However, the results were not as convincing as we had hoped (see our response to comment #6 below). Instead, we followed a different strategy and assessed the distribution of immature (β 1-tubulin negative) and mature MKs (β 1-tubulin positive cells). As expected, mature MKs were bigger in size (Fig. 2D), but the percentage of vessel-associated MKs and the mean distance of non-vessel associated MKs was indistinguishable between immature and mature MKs (Fig. 2E,F). We have now expanded the analysis with regard to bone-associated (BA) and non-bone associated (NBA) MKs (Fig. 2G). Of note, the percentage of β 1-positive MKs was identical for BA and NBA MKs (Fig. 2G). Thus, even though we only have indirect evidence on the distribution of progenitor cells we have clear data that the distribution of immature MKs does not differ from that of mature MKs, arguing against a migration of MKs during their maturation. Nevertheless, we have toned down our statements on p.12 and p.16 to clarify in the manuscript, that our conclusions with regard to the localization of MK progenitors are based on indirect evidence, namely that the vessel-distant bone marrow compartment is very small, omitting the necessity of progenitors to migrate.

4. The authors state that platelet recovery after depletion is due to increased MK turnover. However, they don't measure that directly. They only see increased endomitosis – which suggests increased/faster MK maturation. I think that this is an important point because there needs to be a good explanation for the platelet replenishment in order for this to have physiological relevance. Perhaps this could be addressed with a classical pulse-chase where you look how many labeled MKs are left after certain time points. This would directly address cell turnover and not nuclear content accumulation.

The Reviewer raises a crucial point with respect to chasing specific progenitor cells (or populations) within the bone marrow compartment. We have addressed this concern by two approaches: (1) multiple labeling of MK-specific antibodies in limiting saturations, (2) metabolic DNA-labeling by feeding the nucleoside analogue EdU. Both techniques have technical limitations.

The antibody approach **(1)** did not have sufficient sensitivity. Injection of 15 μ g of the anti-GPIX derivative was adequate to efficiently label all circulating platelets for four days (see Figure to Reviewers 4). However, 24 h after antibody injection all existing MKs could still be stained with the same anti-GPIX derivative (conjugated with a different fluorophore; see Figure to Reviewers 4). Apparently, our "pulse" was not sufficient to saturate all binding sites, despite the fact that platelets were highly positive for the first antibody labeling for three days. Thus, this approach was clearly not feasible for any desired pulse-chase labeling. When we tried 30 μ g anti-GPIX derivative as initial pulse, 24 h later we observed unusual high background fluorescence using LSFM which impeded object recognition and excluded the use of this dosage as a "pulse" to "chase".

In contrast to the antibody labeling, EdU incorporation **(2)** requires that the bioavailability is sufficient for specific labeling of proliferating cells. While this can readily be achieved in cell culture dishes, the bioavailability to mouse bone marrow was reached only 10 days after begin of feeding, when we detected the first EdU-positive lymphocytes in bone marrow sections. Thus, the "pulse" becomes quite broad, with all consequences in our attempt to "chase" it. Therefore, we only modestly state that our EdU-experiment allows to distinguish between a model of a serial "tube" in which all mature MKs are used before new precursors differentiate and take up EdU (first in - first out, all mature MKs become EdU+) and an alternative model in which mature EdU-negative MKs are still present, despite an extended feeding period of at least 10 days (and many 'young' MKs that are EdU-positive). We fully

agree with the Reviewer that this statement needed to be sharpened in the manuscript and we have now revised the paragraph on page 10 and expanded the discussion on p. 16 to make this point clearer.

Figure to Reviewers 4. 15 µg anti-GPIX derivative / mouse are sufficient to label all platelets, but not saturating GPIX on MKs. A, B) Mice received the indicated amount of anti-GPIX-Alexa488 derivative. The percentage of labeled platelets (A) and the mean fluorescence intensity of the Alexa488 emission (B) was determined by flow cytometry. C) Mice received 15 µg anti-GPIX-A647 derivative (red) 24 h after the injection of 30 µg anti-GPIX-Alexa488 derivative (green). Femurs were taken 30 min afterwards and sections subsequently counter-stained with anti-CD105-Alexa 546 (blue) and DAPI (grey). Only double-stained MKs could be detected arguing against this consecutive anti-GPIX labeling strategy as pulse-chase experiment. Scale bar 100 µm.

5. What are the platelet counts in the CXCR and Talin deficiency models? If there is a decrease in platelet count, what accounts for this change since the MK number, distribution, and volumes are unchanged?

We now state in the text that mice lacking Talin1 specifically in platelets and megakaryocytes (PF4-Cre) have normal platelet counts. The same is true for mice that received a short-term treatment with the CXCR4 antagonist AMD3100 (for the data underlying the original Fig. 5, now Fig. 6) vs. vehicle. A constitutive CXCR4-knockout mouse is not viable (Tachibana *et al.*, Nature 1998, 393:591-4; Ma *et al.*, Proc Natl Acad Sci USA 1998, 95:9448-53). We are not aware of MK/platelet specific CXCR4-deficient mice and indeed tried to generate these mice by intercrossing floxed *Cxcr4* mice with PF4-Cre mice. However, in our hands this did not result in absence of CXCR4 in platelets (see Figure to Reviewers 5). We assume that CXCR4 protein expression might be too stable and/or expressed before the PF4-Cre mouse to delete it in MKs/platelets using the PF4-Cre system. A similar observation has been reported for the sphingosine 1-phosphate receptor (*S1pr1f/fi*, Zhang *et al.*, J Exp Med 2012; 209:2165-81).

Figure to Reviewers 5. Platelets of PF4-Cre positive *Cxcr4^{fl/fl}* mice still express CXCR4. Platelets of adult *Cxcr4^{fl/fl}* mice, which either do not express Cre recombinase (Cre negative, lanes 1-3) or express it under control of the platelet-factor 4 promoter (PF4-Cre, lanes 4-6) were tested by Western Blotting for the expression of CXCR4 using the rabbit anti-mouse CXCR4 antibody PA3-305 (Thermo Fisher) and a goat anti-rabbit IgG secondary antibody. GPIIIa served as loading control.

6. Much of the authors' work (including all of the migration data, I believe) was done using CD42a as a marker, which is a marker of mature(ish) MKs. The authors should be careful not to inappropriately extrapolate their observations made on CD42a+ cells to all MK lineage cells, as the MK progenitors are presumably CD42a-. The simultaneous use of CD34 or another similar marker would be useful to positively identify MK progenitors.

We agree with the Reviewer that the direct visualization of MK progenitors would be desirable and pushing the field forward. **The reason why we did not include progenitors into our manuscript is the lack of a suitable surface marker that is specific for early megakaryocytic progenitor cells.** While it would be possible to isolate these cells for flow cytometric analyses which would allow gating and back-gating approaches, this is not possible *in situ*, without losing relevant information on the three-dimensional topology. The limitation of CD34 as a suitable marker protein is that murine megakaryocyte-erythroid progenitors as well as MK progenitors lack uniform expression of CD34, which is in contrast to the situation in human MK progenitors (Yu & Cantor, Platelets and Megakaryocytes – methods in molecular biology, 3rd edition, p. 291-303). Murine MK progenitors have been reported to be CD150⁺, CD41⁺, c-kit⁺, Lin⁻, Sca-1⁻ cells (Pronk *et al.*, Cell Stem Cell 2007; 1:428-42; Nakorn *et al.*, Proc Natl Acad Sci USA 2003; 100:205-10). However, all detected cells co-expressing CD150 and CD41 were already MKs (judged by the presence of CD42 and by an enlarged cell diameter). This could be either due to the low frequency of the MK progenitors (only 0.01% of BM cells are considered to be MK progenitors; Yu & Cantor 2012) or due to the fact that fluorescence microscopy can only highlight stained structures and cannot precisely identify negative staining to categorize cells as it would be possible in flow cytometry. **Consequently, we have toned down our statements regarding MK progenitors and clarified in the manuscript that we have only indirect evidence concerning the localization of MK progenitors in respect to the vasculature / endothelial barrier.**

7. The authors are making an inappropriate presumption that the endosteal niche is devoid of blood vessels. That is old dogma that has long been refuted (PMC4514480, PMC4156024, PMC3984353, PMC4148778 (commenting on PMC3821873), PMC4850557). Ideally, in addition to the distance to the nearest vessel, the distance to the bone should be calculated too.

We agree with the Reviewer that the hematopoietic stem cell community has meanwhile established a model in which endosteal and vascular niches are no longer seen as physically separated. However, this concept has not yet been incorporated into the current models of megakaryopoiesis (see Bluteau *et al.*, *J Thromb Haemost* 2009; 7S1:227-34; Grozovsky *et al.*, *Blood* 2015; 126:1877-84; Malara *et al.*, *Cell Mol Life Sci* 2015; 72:1517-36; Eto & Kunishima, *Blood* 2016; 127:1234-41). Here, we do this for the first time; therefore, we believe that our study will have substantial impact and be relevant to a very broad audience.

Following the Reviewer's suggestion, we have now measured the distance of all MKs to the bone cortex. To this end, we first assessed whether the bone-like structures at the outer linings of femur or sternum sections are indeed positive for collagen I – in line with our previous study (Semeniak *et al.*, *J Cell Sci* 2016; 129:3473-84; see Figure to Reviewers 6A). Moreover, some parts of this structure were positive for osteocalcin (Figure to Reviewers 6B). We noted that these structures displayed a characteristic autofluorescent detection that could be recognized using the established software 'ilastik' (EMBL, Heidelberg, Germany). This approach enabled us **to identify the calcified bone structures in our LSFM images and to perform quantitative analyses. We could clearly show that the mean distance of MKs to the bone differs dramatically between the femur diaphysis on the one hand and sternum, femur epiphysis as well as skull bone on the other hand** (Figure to Reviewers 6C), while the **differences between these four distinct 'bone entities' corresponds well with the total bone-to-bone distances.** We have included these data in the new Figure 4 (see also movies 5-9). Indeed the femur diaphysis is relatively 'poor' in calcified bone, as the bone is present only as an outer shelf surrounding the bone marrow. In contrast, femur epiphysis, sternum and the skull bone have more 'bone invaginations' into the BM (Figure 4, Supplemental Figure 3 and the complementary movies 5-9). Indeed, every space within the bone cortex is completely filled with BM (Figure to the Reviewers 6D and complementary movie 7). Or, in other words, the BM is fully encapsulated by bone cortex. The entire space within this bone capsules is filled with BM (depicted yellow in Figure to the Reviewers 6D), vasculature (red) and MKs (green). Importantly, vessel-to-vessel distance, percentage of MKs with direct vessel contact and MK-to-vessel distance of non-vessel associated MKs is comparable for all types of bone we analyzed (Figure 4). **This indicates that the intrinsic BM architecture and the distribution of MKs within the BM is bone independent and similar between the different types of bone.**

Figure to Reviewers 6. Bone cortex can be identified and used for quantitative analyses. A, B) Cryosections of murine femora were stained for vessels (CD105, red), MKs (CD41, green), collagen I (cyan, only depicted in A) or osteocalcin (blue, depicted in B). Scale bar 100 μm . C) MK distances to bone were determined for different types of bone. D) Different optical z-sections of the LFSM movie 7 illustrate that the entire BM (yellow pseudo-color based on auto-fluorescence signals) is fully encapsulated in bone (grey). Of note, all BM 'caves' contain blood vessels (CD105, red) and MKs (GPIX, green). Scale bar 200 μm .

8. The authors demonstrated quite convincingly that mature(ish) MK's (i.e. CD42a+) have low motility and were distributed around the bone marrow, localized near vessels. However, this is not necessarily in such stark contrast with the model proposed by Rafii. The model (see Fig 6C in his Jan 2004 Nat Med paper) is that HSPCs at the osteoblastic/endosteal niche respond to TPO and generate MK progenitors which then migrate towards the sinusoidal vessels where they undergo maturation, extend proplatelets, and produce platelets into circulation. In my opinion, there is not sufficient evidence in

the current paper to completely invalidate this model, since MK progenitor migration was not assessed, and distance to bone was never measured. That said, recent data, e.g. PMC4850557, has indicated that most HSCs, including quiescent HSCs, are distributed throughout the bone marrow and not near bone, so the current authors' hypothesis is certainly plausible, even likely. This should be addressed.

We thank the Reviewer for this valuable comment. Indeed, we did not intend to contradict Rafii's model on the platelet count stimulating effects of combined SDF1 and FGF4 treatment. Our intention was rather to clarify that mature MKs do not migrate and that the non-vessel associated MKs are distributed randomly within the bone marrow cavity. Moreover, we have clarified in our discussion on p. 13, that the dense blood vessel network severely limits the space for vessel-distant cells. Our data, together with the work of Morrison's group and others showing that HSCs are distributed throughout the BM, makes it likely that the necessity of MK-progenitor cells to migrate is limited as well. **We have rephrased a large section within the discussion section on page 14, now clarifying which of our conclusions are based on direct vs. indirect evidence.**

9. What is the relevance to human platelet production in both health and disease? This should be highlighted and clearly, concisely stated in the discussion and abstract.

We are grateful to this Reviewer comment. As the concentration of platelets in blood is overall tightly regulated: low platelet counts are associated with an increased risk of bleeding and high counts increase the risk of over-shooting platelet activation and thrombotic events that might further embolize. In response to certain disease conditions (i.e. inflammation, bleedings) there is an increased demand in platelets that are rapidly produced by mature MKs, which can immediately release platelets across the endothelial barrier into the blood stream. While there are many drugs available that have an impact on platelet function, the production of platelets can so far not been targeted by drugs. This lack is also due to the fact that the regulatory mechanisms that stimulate or curb terminal platelet biogenesis are largely unknown. Moreover, second generation thrombomimetics affect early MK progenitors with an attenuated kinetic of several days to weeks before platelet counts increase in the periphery. Our study aims to refine the model of where mature MKs are located and how terminal thrombopoiesis can be modulated in healthy humans, and finally, also in patients with a lack or a surplus of platelets. As suggested by the reviewer we have adapted our abstract and included a short paragraph on the human relevance on p.16.

Reviewer #3 (Remarks to the Author):

In this paper Stegner et al. used live two-photon microscopy imaging, computational simulations and immunofluorescence of mouse bone marrow to demonstrate that megakaryocytes do not migrate from periostic to sinusoidal niche but rather accommodate in close contact with sinusoids. Further, platelet depletion, CXCR4 inhibition or talin deficiency seems not to affect megakaryocyte motility or localization in vivo. Their conclusion is that MKs originate and are replenished by sinusoid-resident precursors. However, the concept that MKs are mainly sinusoid-associated is not surprising nor completely original as recently published by the same group (Semeniak et al. *J Cell Sci.* 2016 Sep 15;129(18):3473-84).

Data on MK motility, even though in agreement with previous work by Junt et al. (*Science.* 2007 Sep 21;317(5845):1767-70), would be of interest however some concerns temper enthusiasm:

- Why do the authors expect a high rate of motility in GpIX+ mature MKs, which are already in close contact with sinusoids? Especially in a time window of 30 minutes? Differently from CD41, GpIX is considered a late maturation marker in MKs. Is it possible that immature MKs were excluded from the live analysis within the proposed experimental settings? CD41/ β 1 Tub Double staining in Fig 2C shows the existence of two MK pools. How is GpIX expression distributed among these MK populations?

Despite previous intravital imaging data and mounting evidence of a perisinusoidal localization of hematopoietic stem cells (see Reviewer 2's comment #5), the classical concept of megakaryopoiesis that hematopoietic stem cells mature towards MKs while migrating in response to a gradient from a vessel distant stem cell niche toward the vessel (where terminal platelet biogenesis takes place) is still persisting (see Bluteau *et al.*, *J Thromb Haemost* 2009; 7S1:227-34; Grozovsky *et al.*, *Blood* 2015; 126:1877-84; Malara *et al.*, *Cell Mol Life Sci* 2015; 72:1517-36; Eto & Kunishima, *Blood* 2016; 127:1234-41). Accordingly, we investigated whether MKs are migrating within the bone marrow during their maturation. As a very recent publication of Lefrancais *et al.* (*Nature* 2017; 544:105-9) suggests that a substantial fraction of all circulating platelets are produced in the lungs our current manuscript is timely and of high relevance for the community. Lefrancais *et al.* postulate that whole MKs leave the bone marrow (BM) to enter the capillary bed of the lung, where terminal platelet formation occurs. Even though, we did not focus on the aspect of MK transmigration from the BM towards the lung vasculature, we would like to state that we did not observe transmigration of intact MKs across the BM endothelium. In our 2P-IVM only 2 of over 230 MKs monitored in total were found within the BM vasculature. Likewise, using LSFM imaging and subsequent analysis of intact bones only a tiny fraction ($2 \pm 1.6\%$) of the MKs were detected within the BM vasculature. We have stated this on page 15 of our revised manuscript.

In our previous *J Cell Sci* paper (Semeniak *et al.*) we have addressed the impact of distinct collagen types and their receptors on proplatelet formation. In contrast, this current manuscript assesses MK distribution under steady-state conditions and upon increased platelet demand by using various complementary imaging approaches and has a completely different focus than our previous publication. Moreover, due to the combination of different imaging approaches as well as computational modeling our work technically by far exceeds previous studies, assessing aspects of MK maturation.

As requested by Reviewer 1 (point 1) and suggested here by Reviewer 3, we have extended our intravital microscopy to a prolonged observation time. We have generated 3 h intravital videos and analyzed in total 54 MKs, among those 8 MKs were not in contact of the vasculature. These additional data are now included into the revised version of Figure 1 (see as well Reviewer 1, point 1) and further strengthen our study.

Regarding the expression of CD41 and the GPIb-V-IX complex on maturing MKs, it is indeed generally believed that CD41 precedes the expression of GPIb-IX. However, we could not detect CD41 positive cells which were negative for the GPIb β or GPIX (see new Supplementary Figure 1); neither using anti-CD41 antibodies nor using the CD41-YFP knockin mouse (in which $34.2 \pm 7.2\%$ of GPIX-positive MKs were YFP-positive, but no YFP-positive, GPIX-negative cells could be detected – data not shown). Consequently, no difference with regard to the distribution of β 1-tubulin positive MKs could be observed between CD41 positive and GPIX-positive MK pools.

- MK migration in Figure 1 was measured in 37 cells by 2P-IVM. Which cells were selected for these measurements? Were they exclusively vessel-associated MKs or randomly distributed cells?

We have included all MKs that did not release pro-platelets during the observation period; we have highlighted this in the method section on p. 22. In line with the general distribution of MKs, the majority of these MKs was in direct contact with the vessel, and only 8 out of 54 observed MKs were not in contact of the vasculature. Importantly, the MSD analysis does not differ between vessel-associated and non-vessel associated MKs, indicating that MKs are sessile independently of their spatial distribution. These data are now included into the revised version of Figure 1.

Were MKs associated with periostic osteoblasts included in the analysis? Why wasn't a comparison carried out between periostic-associated MKs vs sinusoids-associated MKs in terms of instantaneous migration? In order to exclude a potential MK origin from the endosteal niche, this site should be analyzed as well. Periostic niche is present only in the immunofluorescence of Fig. 2 of sternum and femora. Is cranial BM suitable for analyzing endosteal-MK contacts to a similar extent of BM femur/sternum? This topic is only discussed briefly in the discussion section.

We are very grateful for this comment, as we have so far neglected the bone site in the first submission of the manuscript. We have now taken effort to address the concern raised by this Reviewer. So far, there is – to the best of our knowledge – no report in the literature that compared cranial BM to that of femur or sternum, despite being frequently used for intravital imaging. One reason might be that it is challenging to perform consecutive cryo-sections of cranial bone for subsequent analyses. Instead, **we performed an analog analysis using LSFM of cranial BM ourselves**. Interestingly, the bone-to-bone, vessel-to-vessel or vessel-to-MK distances were comparable between cranial BM, sternum and femur epiphysis (see new Figure 4 and new Supplementary Figure 3).

To address the Reviewers request with regard to “endosteal-MK contacts” **we have determined the bone-to-bone distances and relevant MK parameters for the different types of bone**. This information is now included as new Figure 4 and part of new paragraph within the results section (p. 7-8) and the discussion (p. 13). Moreover, we have chosen (1) femur diaphysis, as this long bone has the largest bone-to-bone distance, and (2) sternum as a characteristic bone with overall smaller bone-to-bone distances and have compared bone-associated and non-bone-associated MKs. To this end, we defined a 100 μ m zone close to the bone structures as ‘bone-associated’ (BA; based on the assumption that a MK can have diameters of up to 50 μ m and two diameters could be a reasonable distance) and the remaining BM as non-bone associated (NBA; see Figure 4). Following this subdivision, we could not observe any differences between BA and NBA MKs regarding the average vessel association, MK distances to vessels (of non-vessel-associated MKs) or MK diameters (see Figure 4). In conclusion, it appears to be rather unlikely that a ‘classical’ endosteal niche exists for MKs.

The proposed differentiation between the migration of bone-associated MKs vs. vessel-associated MKs, is problematic as the entire space within the bone 'capsules' is fully vascularized (see our response to comment 7 of Reviewer 2; Fig. 4, Supplemental Fig. 3, Figure to Reviewers 6). Thus, there are virtually no 'periostic' MKs that would not be close to sinusoids. In addition, a differentiation between bone-associated (BA) and non-bone-associated (NBA) MKs is not possible for the skull bone marrow, as the bone-to-bone distance of less than 200 μm (see Fig. 4B). Thus, 2P-IVM of NBA vs. BA MKs is impossible, as almost all MKs are BA. As an alternative approach, we assessed the percentage of β 1-tubulin positive CD41-cells in bone-associated (BA; within 100 μm distance from the bone cortex) and non-bone associated (NBA) MKs in femur-sections. However, we did not observe any differences. This is now included in the text on p. 6 and in Fig. 2G. Moreover, we have included these data in the discussion section on p. 13, as it indicates that the fraction of mature MKs (= β 1-tubulin positive cells) does not differ between BA and NBA MKs arguing against migration of determined MKs during their maturation.

In general, the conclusions of this manuscript are speculative with respect to the data presented.

Major Concerns:

A major point of concern is the use of an anti-GpIX antibody as the live tracking system for MK visualization. As stated by authors, this labeling scheme was required to overcome the limitations of the CD41-YFP mouse published by Junt et al. 2007 in Science. However, details of the source of the GpIX antibody are not provided (e.g., clone, epitope, manufacturer) and several missing controls compromise the major premises of this paper. For example, experiments demonstrating negligible effects of this GpIX antibody on MK function (PPF, Adhesion/migration) *in vitro* should be provided. What are the effects of GpIX ab injection on mice peripheral blood parameters?

We have used a rat anti-GPIX-derivative (derived from clone p0p6) (Nieswandt *et al.*, Blood 2000, 96:2520-7). The antibody is generated by a hybridoma cell line and is constantly purified and derivatised in Dr. Nieswandt's laboratory (Nieswandt *et al.*, Blood 2000; 96:2520-7). The epitope recognized by this antibody is well characterized as the extracellular domain of GPIX. Application of this anti-GPIX derivative does not affect platelet production, neither *in vitro* (see new Supplementary Figure 1), nor *in vivo* as 1.5 $\mu\text{g/g}$ body weight (= more than twice the amount of antibody used for 2PIVM) did not affect platelet counts on day 1 or day 3 after injection compared to vehicle-treated mice (see Figure to Reviewers 3).

- Why weren't experiments in mice with CXCR4 inhibition or talin deficiency performed by 2P-IVM? In this regard the authors should explain why MK distribution in mice with CXCR4 inhibition or talin deficiency was assessed by immunofluorescence with an anti-CD41 antibody that recognizes both immature and mature cells (and not the same GpIX ab)?

The disadvantage of 2P-IVM is the relatively small field of view, which does not allow for a comprehensive overview of the total MK distribution within a bone. In contrast, femoral sections are more frequently used to analyze number and localization of MKs in various KO mice (e.g. Bruns *et al.*, Nat Med 2014; 11:1315-20; Meinders *et al.*, Blood 2015; 125:1957-67; Tamura *et al.*, Blood 2016; 13:1701-10; Semeniak *et al.*, J Cell Sci 2016; 129:3473-84). To allow direct comparison with previous studies by us and others, we have decided to analyze cryosections of femora. Likewise, expression of CD41 is the 'gold standard' for MK labeling which we have used for all cryosections (also those depicted in Figure 2) of the initial manuscript. We only used an anti-GPIX derivative for LSFM or intravital

microscopy as this reagent does not affect platelet counts or function and has no detectable effect on production and can therefore be administered *in vivo* (see our response to Reviewer 2, point 1). As depicted in our new Supplementary Figure 1 there are virtually no GPIX-negative CD41-positive cells detectable in cryo-sections of murine femora, making it very unlikely that the use of anti-CD41 or anti-GPIX antibodies has any significant effect on the results of the performed experiments.

- What concentration of SDF-1 was used? Was the protocol different from that of Niswander et al. Blood. 2014 Jul 10;124(2):277-86? The authors should explain or at least discuss discrepancies with this work. Was the peripheral blood platelet count affected by AMD3100 or SDF-1 treatment?

We have used 16 µg/kg bodyweight of SDF-1, which reflects 400 ng SDF-1 for a 25 g mouse, identical to the doses used by Niswander *et al.*. We apologize that we had provided this information only under “antibodies and reagents” in the previous version of our manuscript. We have now added this information to the figure legend as well. Niswander *et al.* reported a small increase (<20%) in peripheral platelet counts on day 1 after SDF-1 treatment. However, the investigators of this study did astonishingly find rather low platelet counts in control mice (only 430/nL as stated in Figure 1 of Niswander *et al.*), which is far below the 1000/nL that are typically reported for C57BL/6 mice). We found (910 ± 86) platelets per nL in our vehicle-treated animals. 1 h after SDF-1 injection of 16 µg/kg bodyweight we observed a slightly elevated platelet counts (1039 ± 113 platelets/nl blood) – which was comparable to the reported effects of SDF-1 treatment of Niswander *et al.* (14% vs. 18% increase in platelet counts). However, after 24 h we had platelet counts of 821 ± 139 platelets/nl. This does not necessarily contradict the report of Niswander *et al.*, as they observed a transient effect of SDF-1 treatment, which they suggested to trigger platelet release from already mature MKs (Niswander *et al.*; Blood 2014, 124:277-86).

In agreement with the SDF-1 results, we saw a small decrease in platelet counts 1 h after AMD3100-treatment (768 ± 72 platelets per nl blood vs. 977 ± 56 platelets/nl blood in vehicle treated mice). However, 24 h after AMD3100 treatment platelet counts were restored (987 ± 185 platelets/nl). We have now mentioned the effects of SDF-1 and AMD3100-treatment on platelet counts in the text on p. 11.

- Description of results are too speculative and conclusions not fully supported by experiments. For example, the increased rate of cell ploidy after platelet depletion is described as “reflecting the increased MK turnover due to elevated platelet demand” but this finding was not experimentally tested. “MK turnover” or “MK replenishment by precursors” are terms found within the manuscript but these concerns were not experimentally assessed.

We appreciate the Reviewer’s point and have clarified within the manuscript which of our conclusions are based on direct evidence and which is rather indirect evidence considering our data and previous work of others (see p. 12 and 16). We are confident that this increased the clarity of our manuscript.

Minor Comments:

- Figure 6A is not clear.

We have changed the shading so that it should now be clearer that in Figure 7A (previously Figure 6A) the entire MK population (including vessel-associated and non-vessel associated MKs) is depicted.

- Is there a reason for injecting a 1000-fold more concentration of GpIX ab with respect to previously published paper by the same group (0.6 μ g/g mice in Bender et al. Nat Commun. 2014 Sep 4;5:4746. doi:10.1038/ncomms7507 vs 0.6mg/g mice in this paper)?

We apologize for this typing error. Apparently, we have replaced μ g by mg in this entire paragraph during the final formatting of our manuscript. We did use exactly the same concentration of anti-GPIX derivative that we have previously used.

REVIEWERS' COMMENTS:

Reviewer #1 (Remarks to the Author):

The authors have addressed all my concerns. The manuscript is now both clearer and more balanced. My only last request is to include Figure to Reviewers 2 (and 3 and 6) in the manuscript, even if just as supplementary figures, as they are important controls that strengthen the manuscript and should be shared with the readers.

Reviewer #2 (Remarks to the Author):

The authors have done a very thorough job addressing our concerns. In addition, I think they also did an exceptional job addressing the other reviewer's concerns. The improvements are really evident. The data is of very high quality, the images are beyond impressive, the findings are novel and very convincing, and the work will make a big impact in the field. I actually think it should be in a higher level journal at this point.

Reviewer #3 (Remarks to the Author):

The authors have addressed all my concerns regarding the experimental aspects of the study. However, I suggest one major revision to improve the novelty of the data presented and render the manuscript suitable for publication in Nature Communications. To strengthen the message of the slow migration as a hallmark of MKs within BM, the authors should measure by IVM, under the same experimental conditions, the migration rate of a bone marrow lineage other than Mk (e.g. GpIX ab in combination with a plasma cell or monocyte marker).

Reviewers' comments:

Reviewer #1 (Remarks to the Author):

The authors have addressed all my concerns. The manuscript is now both clearer and more balanced.

My only last request is to include Figure to Reviewers 2 (and 3 and 6) in the manuscript, even if just as supplementary figures, as they are important controls that strengthen the manuscript and should be shared with the readers.

We thank the Reviewer for the positive feedback. As recommended we have included the three figures into the supplementary files.

Reviewer #2 (Remarks to the Author):

The authors have done a very thorough job addressing our concerns. In addition, I think they also did an exceptional job addressing the other reviewer's concerns. The improvements are really evident. The data is of very high quality, the images are beyond impressive, the findings are novel and very convincing, and the work will make a big impact in the field. I actually think it should be in a higher level journal at this point.

We appreciate the Reviewer's encouraging feedback of our work!

Reviewer #3 (Remarks to the Author):

The authors have addressed all my concerns regarding the experimental aspects of the study. However, I suggest one major revision to improve the novelty of the data presented and render the manuscript suitable for publication in Nature Communications. To strengthen the message of the slow migration as a hallmark of MKs within BM, the authors should measure by IVM, under the same experimental conditions, the migration rate of a bone marrow lineage other than Mk (e.g. GpIX ab in combination with a plasma cell or monocyte marker).

We are grateful that the reviewer considers our manuscript to be now suitable for publication in Nature Communications. We do understand the reviewer's point of having a second bone marrow lineage tracked in addition to MKs. However, a comparison of the migration of MKs and B cells has been published already in Junt et al. (Science 2007; 317:1767-70), which revealed that MK cells have significantly lower motility as compared to B cells. Therefore, we do not think that repeating these experiments would significantly strengthen our manuscript further.